# Cell crowding activates pro-invasive mechanotransduction pathway in high-grade DCIS via TRPV4 inhibition and cell volume reduction

Xiangning Bu[1], Nathanael Ashby[1], Teresa Vitali[1], Sulgi Lee[1], Ananya Gottumukkala[1,2], Kangsun Yun[1], Sana Tabbara[3†], Patricia Latham[3], Christine Teal[4], Inhee Chung[1,5]*

[1]Department of Anatomy and Cell Biology, School of Medicine and Health Sciences, George Washington University, Washington, DC, United States; [2]Thomas Jefferson High School for Science and Technology, Alexandria, United States; [3]Department of Pathology, George Washington Medical Faculty Associates, Washington, DC, United States; [4]Department of Surgery, George Washington Medical Faculty Associates, Washington, DC, United States; [5]Department of Biomedical Engineering, GW School of Engineering and Applied Science, George Washington University, Washington, DC, United States

*For correspondence:
inheec@gwu.edu

Present address: †Department of Pathology, H. Lee Moffitt Cancer Center and Department of Oncologic Sciences, University of South Florida Morsani College of Medicine, Tampa, United States

## eLife Assessment

This **fundamental** study provides **compelling** evidence that TRPV4 plays a crucial role in mechanical sensing during cancer cell transition from non-invasive to invasive states, and offers novel insights into metastasis. By employing multiple experimental approaches, including pharmacological and genetic manipulation, as well as advanced imaging techniques, the authors demonstrate a strong correlation between TRPV4 dynamics, calcium homeostasis, and cell volume plasticity. The findings significantly enhance our understanding of mechanotransduction in cancer and present TRPV4 as a promising therapeutic target for inhibiting metastasis.

**Abstract** Cell crowding is a common microenvironmental factor influencing various disease processes, but its role in promoting cell invasiveness remains unclear. This study investigates the biomechanical changes induced by cell crowding, focusing on pro-invasive cell volume reduction in ductal carcinoma in situ (DCIS). Crowding specifically enhanced invasiveness in high-grade DCIS cells through significant volume reduction compared to hyperplasia-mimicking or normal cells. Mass spectrometry revealed that crowding selectively relocated ion channels, including TRPV4, to the plasma membrane in high-grade DCIS cells. TRPV4 inhibition triggered by crowding decreased intracellular calcium levels, reduced cell volume, and increased invasion and motility. During this process, TRPV4 membrane relocation primed the channel for later activation, compensating for calcium loss. Analyses of patient-derived breast cancer tissues confirmed that plasma membrane-associated TRPV4 is specific to high-grade DCIS and indicates the presence of a pro-invasive cell volume reduction mechanotransduction pathway. Hyperosmotic conditions and pharmacologic TRPV4 inhibition mimicked crowding-induced effects, while TRPV4 activation reversed them. Silencing TRPV4 diminished mechanotransduction in high-grade DCIS cells, reducing calcium depletion, volume reduction, and motility. This study uncovers a novel pro-invasive mechanotransduction pathway driven by cell crowding and identifies TRPV4 as a potential biomarker for predicting invasion risk in DCIS patients.

**eLife digest** Ductal carcinoma in situ (known as DCIS) is an early form of breast cancer that develops in the milk ducts. It is non-invasive, which means that it does not spread into the surrounding breast tissue. Despite this, if left untreated, DCIS can develop into an invasive cancer that spreads to nearby tissues and has the potential to spread to other parts of the body.

It is difficult to predict which DCIS cases will develop into invasive breast cancers. Although DCIS cells can be graded according to how abnormal their appearance is, with high-grade being the most abnormal, this classification does not directly predict their invasive potential. Therefore, there is a need to develop new ways to better predict which DCIS cases are at risk of progressing to invasive cancers.

Bu et al. aimed to investigate whether cell crowding impacts how likely a DCIS cell is to become invasive. The rapid proliferation of cancer cells in confined spaces means that they often become crowded together, causing mechanical stress that can change their features and behaviour.

Bu et al. probed the effects of cell crowding on different types of breast cancer cells, including healthy breast cells, non-invasive DCIS cells of different grades and invasive breast cancer cells. The experiments revealed that while cell crowding does not drive low-grade DCIS and non-cancerous cells to become invasive, it can promote invasiveness in high-grade DCIS cells. This is due to inhibition of proteins that allow calcium ions to pass into the cells. In particular, this inhibition of a protein known as TRPV4 reduces the number of calcium ions inside the cells, which makes the cells smaller and able to move more efficiently. Cell crowding also caused TRPV4 to move from the centre of the high-grade DCIS cells to the cell membrane.

Taken together, the findings reveal that different types of cancer cells respond differently to mechanical stress and identify cell crowding as a factor that can cause high-grade DCIS cells to become invasive. While further investigation is needed, the findings also suggest the location of TRPV4 as a potential marker of DCIS cells that are more likely to become invasive. In the future, markers of potential invasiveness could be used to ensure that cancer treatments are targeted specifically at patients with a greater risk of developing invasive breast cancer, preventing overtreatment of lower risk cases.

## Introduction

The complex interplay between cellular mechanics and invasive behaviors is crucial to understanding various physiological and pathological processes, such as wound healing (*Kuehlmann et al., 2020*) and disease progression (*Broders-Bondon et al., 2018*; *Hall et al., 2021*). Numerous studies show that these processes are mediated by mechanotransduction, whereby cells translate mechanical stimuli into cellular activity (*Broders-Bondon et al., 2018*; *Moore et al., 2010*; *Jansen et al., 2017*; *Jaalouk and Lammerding, 2009*; *Lee et al., 2019a*). While mechanotransduction is traditionally studied concerning fluid stress (*Stylianopoulos et al., 2013*; *Polacheck et al., 2014*), matrix stiffness (*Moore et al., 2010*; *Galbraith et al., 2002*), and other biomechanical changes such as osmotic stress (*Burg et al., 2007*; *Finan and Guilak, 2010*), the role of cell crowding, characterized by increased cell density and spatial constraints, is relatively less explored. A recent report shows that cell crowding plays a role in facilitating wound closure and repair by enhancing cell proliferation (*Franco et al., 2019*). As tissues develop, repair, or undergo pathological transformation, cell crowding becomes common (*Neurohr and Amon, 2020*; *Fan and Meyer, 2021*). This challenges individual cells, forcing them to perceive and respond to the mechanical constraints of a crowded environment. Our study used human breast cell line model systems to describe a novel mechanotransduction pathway triggered by cell crowding that induces invasiveness into surrounding tissues. Interestingly, this pathway exhibited unique selectivity, as it was specifically associated with a type of non-invasive cancer pathology and was not present in lower-grade or less aggressive pathologies. This suggests that not all cells possess the ability to translate mechanical stimuli, such as cell crowding, into cell invasiveness.

To assess the role of cell crowding in cell invasiveness, we chose in vitro cell lines linked to different pathological states that reflect cell crowding conditions in vivo, including atypical ductal hyperplasia (ADH) (*Kader et al., 2018*; *Pinder and Ellis, 2003*) and ductal carcinoma in situ (DCIS) (*Böcker, 1997*; *Pinder and Ellis, 2003*). ADH is an intraductal clonal epithelial cell proliferative lesion (*Pinder and Ellis, 2003*) and represents an intermediate step between normal breast tissue and in situ carcinomas

(*Dupont and Page, 1985*). ADH is associated with high risk as it is reported that an ADH diagnosis is associated with a fivefold increased risk of breast cancer (*Kader et al., 2018*). DCIS is a non-invasive form of cancer characterized by proliferating malignant epithelial cells (*Böcker, 1997*; *Pinder and Ellis, 2003*). Unlike ADH, DCIS is considered a precursor to invasive breast cancer (*van Seijen et al., 2019*). However, the mechanism of how DCIS transitions to invasive cancer is not well understood and therefore, there is currently no reliable and robust method to differentiate which DCIS cells are at high risk of becoming invasive (*Pang et al., 2016*; *Zhou et al., 2009*).

Both ADH and DCIS conditions can potentially expose cells to crowding in vivo. However, how ADH and DCIS cells respond to such changes in cell density remains unknown. Our study revealed that cell crowding selectively triggered a pro-invasive mechanotransduction program in a specific type of DCIS cell line associated with high-grade pathology (*Shekhar et al., 2008*; *Miller et al., 2000*). The mechanotransduction program induced by cell crowding involved the inhibition of ion channels, such as transient receptor potential vanilloid 4 (TRPV4) (*White et al., 2016*), a calcium-permeable ion channel, as identified from our mass spectrometry assay. This inhibition decreased intracellular calcium levels and inactivated TRPV4 and other ion channels, prompting their relocation to the plasma membrane. The inhibition also induced reduced cell volume and cortical stiffening, thereby promoting cell invasiveness and motility.

## Results

### Cell crowding enhances invasiveness in high-grade DCIS cells

Cell crowding reflects a common condition in tumor microenvironments for ADH and DCIS, arising from aberrant cell proliferation within spatially constrained intraductal spaces. We examined the influence of this prevalent environmental factor on the invasiveness of these cells in vitro. To conduct this investigation, we assembled a panel of cell lines derived from the normal breast epithelial cell line MCF10A, including its H-RAS mutation-driven derivatives associated with various pathologies (*So et al., 2012*; *Miller, 2000*): MCF10AT1, MCF10DCIS.com, and invasive cells MCF10CA1a. MCF10AT1 resembles ADH (*Dawson et al., 1996*), MCF10DCIS.com mimics high-grade DCIS (*Miller et al., 2000*), and MCF10CA1a represents a malignant invasive cancer that was observed to form metastatic lesions in a mouse xenograft (*Santner et al., 2001*). The current classification of DCIS relies on histological factors such as cell growth patterns and cytonuclear features (*Holland et al., 1994*; *Allred, 2010*). Comedo-DCIS is a histologic subtype, which is characterized by apoptotic cell death, representing a high-grade DCIS with higher invasive potential than those of lower-grades (*Shekhar et al., 2008*). MCF10DCIS.com forms comedo-type DCIS lesions that can spontaneously transition to invasive cancer when xenografted (*Miller et al., 2000*).

To compare cell invasiveness under normal cell density and cell crowding conditions in vitro, we opted for a modified 2D matrix degradation assay. This approach allowed us to quantify overall cell invasiveness by normalizing it with the total cell number, thereby accounting for differences in cell densities. We chose this method over transwell, 3D Matrigel, or spheroid assays, where quantifying cell invasiveness as a function of cell density is challenging. By modifying an existing collagen-crosslinked polyacrylamide hydrogel matrix-based invasion assay (*Denisin and Pruitt, 2016*; *Fischer et al., 2012*), we could determine the invasive cell fraction out of the total cell population by fluorescence imaging of the invasion gel bed and cell nuclei through low magnification (4 X) imaging (*Figure 1A*, *Figure 1—figure supplement 1* for a detailed procedure) when cells were at normal density versus under cell crowding conditions.

We observed alterations in live cell morphology and invasiveness as they progressed from normal density (ND: 40–70%) to full confluence (100%) and beyond. Our goal was to determine a distinct time window, termed overconfluence (OC), where cell crowding-induced changes level off. The OC time window was achieved in cells cultured for an additional 5–10 d after reaching 100% confluence, with the growth medium replaced twice per day to prevent acidification. We confirmed that the cells remained largely viable under crowding conditions, with minimal cell death (~1.58%) similar to that observed in ND cells (~0.85%) (*Figure 1—figure supplement 2A–B*). The schematic of the timeline to achieve the OC conditions, and low-magnification (4 X) bright-field images at different time points from the ND to OC conditions are shown in *Figure 1—figure supplement 3A–B*. The time-dependent equilibration of cell invasiveness after the cells reached confluence is demonstrated

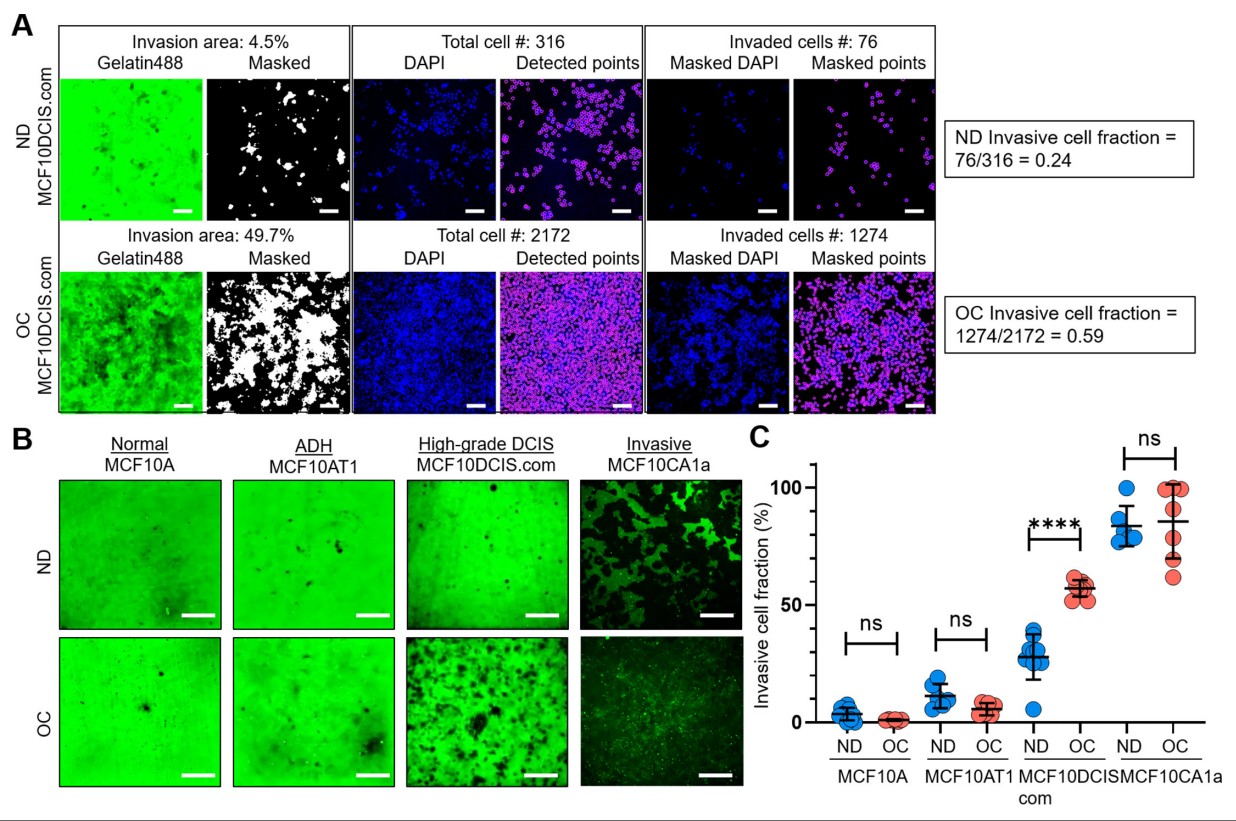

**Figure 1.** Cell crowding selectively increases invasiveness in high-grade ductal carcinoma in situ (DCIS) cells. (**A**) We used a collagen-crosslinked polyacrylamide hydrogel matrix-based invasion assay to assess the effect of cell crowding on cell invasiveness. Representative images show gelatin-Alexa488 conjugates, where dark areas in the green (Gelatin488) background indicate cell invasion through degradation, and DAPI staining marks cell locations (blue; DAPI) in a 2 d invasion assay of MCF10DCIS.com cells under normal density (ND; upper panel) and overconfluent (OC; lower panel) conditions. 'Masked' images are thresholded to produce positive masks applied to the 'DAPI' images. Individual cell locations detected in 'DAPI' images are marked with purple circles in 'Detected points' images. The total number of cells within the field of view is counted from these points. By overlaying the mask and DAPI images, 'Masked DAPI' images are obtained, and invaded cells are detected and represented by purple circles in 'Masked points' images. The invasive cell fraction is calculated by the ratio of the number of invaded cells to the total number of cells (0.24 for ND and 0.59 for OC MCF10DCIS.com cells). These data show that cell invasiveness is enhanced by cell crowding. Scale bar = 100 µm. (**B**) Comparison of 'Gelatin488' images of MCF10A (normal breast epithelial cells), MCF10AT1 (ADH-mimicking cells), MCF10DCIS.com (high-grade DCIS mimic), and MCF10CA1a (invasive breast cancer cells) between ND and OC conditions. MCF10DCIS.com cell invasion is significantly higher under cell crowding than under ND conditions. (**C**) Invasive cell fractions of these cells between ND (blue circles) and OC (red circles) conditions are compared, showing that cell crowding-induced increases in invasiveness are notable only in MCF10DCIS.com cells. The number of cell invasion analyses was as follows: MCF10A (ND: 10; OC: 7), MCF10AT1 (ND: 6; OC: 6), MCF10DCIS.com (ND: 9; OC: 9), and MCF10CA1a (ND: 6; OC: 7). We used the two-tailed Mann-Whitney U test, a nonparametric and unpaired statistical method, to compare differences between groups. ****p<0.0001, ***p<0.001, **p<0.01, *p<0.05, ns: p>0.05, throughout the manuscript.

The online version of this article includes the following source data and figure supplement(s) for figure 1:

**Source data 1.** Original data corresponding to panels A and C.

**Figure supplement 1.** Quantifying the Invasive cell fraction using a 2D polyacrylamide hydrogel-based invasion assay.

**Figure supplement 2.** MCF10DCIS.com cells exhibited comparable viability under overconfluence (OC) conditions to those under normal density (ND) conditions.

**Figure supplement 2—source data 1.** Original data for panel A.

**Figure supplement 3.** Time window for cell crowding conditions in vitro.

**Figure supplement 3—source data 1.** Original data corresponding to panel D.

**Figure supplement 3—source data 2.** Original data corresponding to panel E.

**Figure supplement 3—source data 3.** Original data corresponding to panel F.

in *Figure 1—figure supplement 3C–D* using our quantifiable collagen-crosslinked polyacrylamide hydrogel matrix-based invasion assay. Cell crowding mechanosensing significantly increased the invasive cell fraction of MCF10DCIS.com cells from ~24 (ND cells) to 59% (OC cells) (*Figure 1A*). The enhanced invasiveness of MCF10DCIS.com cells under cell crowding was largely reversible. When OC cells were reseeded at normal density for invasion assays, their invasive cell fraction decreased to approximately 15%, slightly lower (p=0.012) than the initial value of around 24% (*Figure 1—figure supplement 3C and E*). In contrast, normal breast epithelial cells (MCF10A) and the ADH-mimicking MCF10AT1 cells did not display invasiveness under ND conditions or enhanced invasiveness under OC conditions, indicating insensitivity to cell crowding (*Figure 1B–C*).

The invasive cell fraction of ND MCF10CA1a cells was already ~80%, and the additional increase in invasiveness due to cell crowding was not readily discernible, with a slight increase to ~82% under crowded conditions (*Figure 1B–C*). These data suggest a striking and selective mechanosensing effect of cell crowding on cell invasiveness in MCF10DCIS.com cells. To ensure that acidification did not affect the invasiveness of MCF10DCIS.com cells despite the frequent replacement of the cell growth medium, we incubated ND cells for two days with acidified medium from cultures of OC MCF10DCIS.com cells. We observed that medium acidity did not alter cell invasiveness (*Figure 1—figure supplement 3F*), reinforcing that the increased invasiveness under OC conditions was induced by cell crowding.

## Cell crowding reduces cell volume and stiffens MCF10DCIS.com cells

As cells became crowded, we observed a reduction in cell size. Previous reports indicate an inverse relationship between cell volume and cortical stiffness (*Tzur et al., 2009*; *Guo et al., 2017*), leading us to hypothesize that reduced cell volume would be accompanied by increased cortical stiffness. Research on glioma cell invasion underscores the critical role of hydrodynamic cell volume changes in penetration into surroundings (*Cuddapah et al., 2014*; *Watkins and Sontheimer, 2011*), suggesting that significant cell volume reduction facilitates cell invasion. Additionally, increased cortical stiffness is known to help cells overcome the physical barriers of the dense extracellular matrix (*Angstadt et al., 2022*; *Gudjonsson et al., 2005*; *Barai et al., 2021*). We thus investigated whether cell crowding selectively induces pro-invasive cell volume reduction and cell stiffening in MCF10DCIS.com cells, thereby priming them for invasion.

To measure the cell volume of individual cells, we used confocal microscopy with a 60 X oil immersion objective to obtain z-stack images of live or fixed cells stably expressing red fluorescent protein (RFP) (*Shcherbakova et al., 2016*). Using a nanoindenter attached to the confocal microscope (*Figure 2—figure supplement 1A*), we extracted Young's modulus of live individual cells to assess changes in cortical stiffness. Cell crowding significantly reduced cell volume (*Figure 2A and B*) and increased cortical stiffness (*Figure 2C*) in both MCF10DCIS.com and MCF10A cells. However, the changes in volume and stiffness from ND to OC conditions were more pronounced in MCF10DCIS.com cells (*Figure 2—figure supplement 1B, C*). The cell crowding-induced alterations in cell volume and stiffness were negligible in MCF10AT1 cells (*Figure 2A–C*). In MCF10CA1a cells, cell crowding led to an increase in stiffness, but a decrease in volume was not evident. This was expected because the cell volume was already too small to be accurately measured with the available resolution (*Figure 2A–C*). The notable plasticity in cell volume and stiffness changes observed in MCF10DCIS.com cells in response to cell crowding potentially underscores the critical link between this plasticity and mechanosensitive increases in cell invasiveness. We assessed cell volume changes only as an effector event of cell crowding, without measuring cell stiffness, because cell volume reflects the mechanical properties of the entire cell, while Young's modulus can vary depending on the location of indentations in the plasma membrane (*Radmacher, 2002*; *Guilak et al., 2000*). The observed cell volume changes by cell crowding (*Figure 2A*) depended on the ND cell volume, as OC cell volumes under crowding conditions were comparable across all cells. Consequently, the ND cell volume and the volume changes from ND to OC conditions exhibited an approximately linear relationship ($R^2$~0.97; *Figure 2—figure supplement 1D*). This finding suggests that the ND cell volume could serve as an indicator of relative cell volume plasticity.

To validate the pro-invasive nature of cell volume reduction in MCF10DCIS.com cells, we examined the effect of hyperosmotic cell volume reduction in ND cells using polyethylene glycol (PEG) 300 for 15 min. The cell volume reduction effects of this treatment were previously described (*Guo et al.,*

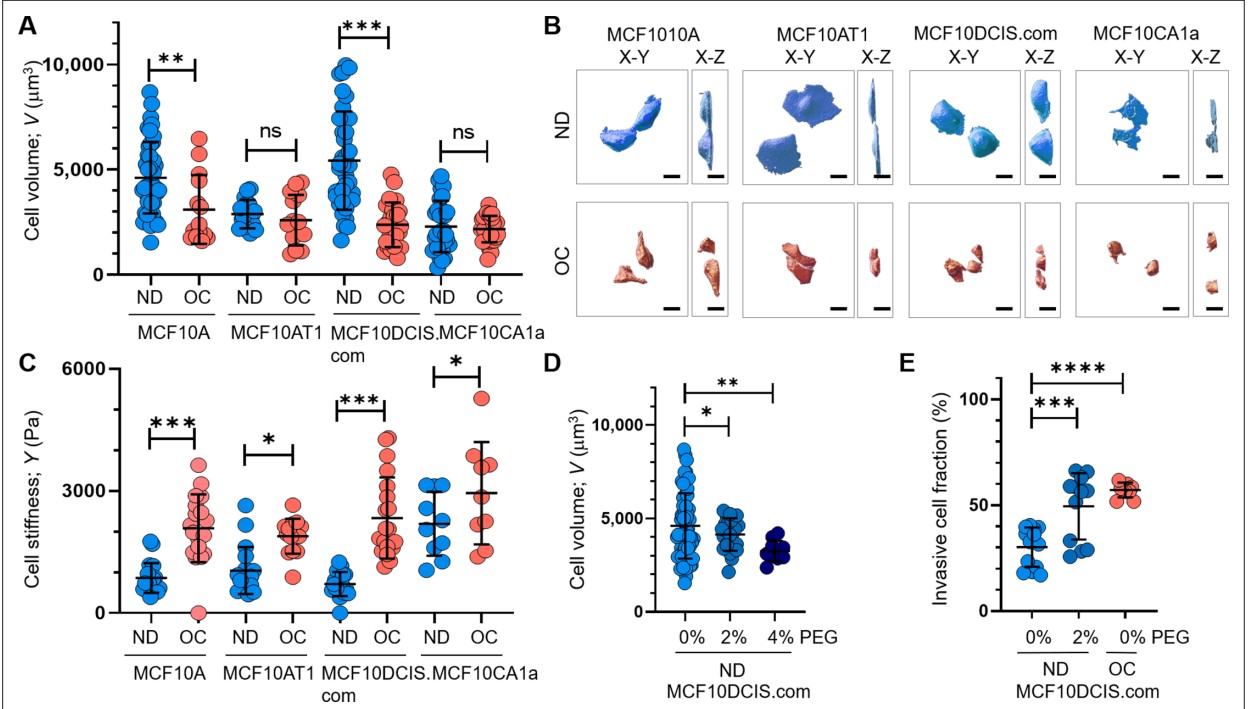

**Figure 2.** Cell crowding induces significant cell volume reduction and stiffening in high-grade ductal carcinoma in situ (DCIS) cells. Cell volume (*V*; mean and SD) differences between normal density (ND) (blue circles) and overconfluence (OC) (red circles) conditions of MCF10A, MCF10AT1, MCF10DCIS. com, and MCF10CA1a cells are plotted. The high-grade DCIS cell mimic, MCF10DCIS.com, shows a large volume reduction due to cell crowding. The number of single-cell volume analyses (technical replicates merged from three independent experimental repeats) was as follows: MCF10A (ND: 44; OC: 14), MCF10AT1 (ND: 16; OC: 16), MCF10DCIS.com (ND: 38; OC: 24), and MCF10CA1a (ND: 29; OC: 31). (**B**) Representative confocal microscopy images of RFP-coexpressing cells of the four cell types in ND and OC conditions. The images include x-y (left) and x-z (right) views, with scale bar = 10 μm. The large volume reduction of MCF10DCIS.com cells is evident. (**C**) Plots showing changes in cortical stiffness (mean and SD) measured by Young's modulus (*Y*) using a nanoindenter, displaying significant cell stiffening of MCF10DCIS.com cells due to cell crowding. The number of single-cell stiffness measurements (technical replicates merged from two independent experimental repeats) was as follows: MCF10A (ND: 21; OC: 19), MCF10AT1 (ND: 19; OC: 14), MCF10DCIS.com (ND: 19; OC: 21), and MCF10CA1a (ND: 11; OC: 10). (**D**) Hyperosmotic conditions induced by PEG 300 treatment (light blue and darker blue circles for untreated and 2% PEG 300=74.4 mOsm/Kg, respectively; navy circles for 4% PEG 300=148.8 mOsm/kg) lead to dose-dependent cell volume reduction. The number of single-cell volume analyses (technical replicates merged from three independent experimental repeats) was as follows: MCF10DCIS.com (ND control: 62; ND +2% PEG: 23; ND +4% PEG: 10). (**E**) Treatment with 2% PEG 300 (darker blue circles) for 2 d significantly increased the invasiveness (mean and SD) of MCF10DCIS.com cells, similar to the OC case (red circles). The number of cell invasion analyses (technical replicates merged from two independent experimental repeats) was as follows: MCF10DCIS.com (ND control: 13; ND +4% PEG: 12; OC: 10). ****p<0.0001, ***p<0.001, **p<0.01, *p<0.05, ns: p>0.05.

The online version of this article includes the following source data and figure supplement(s) for figure 2:

**Source data 1.** Original data corresponding to panel A.

**Source data 2.** Original data corresponding to panel C.

**Source data 3.** Original data corresponding to panel D.

**Source data 4.** Original data corresponding to panel E.

**Figure supplement 1.** Increased invasiveness of MCF10DCIS.com cells correlates with cell volume plasticity.

**Figure supplement 1—source data 1.** Original data corresponding to panel B.

**Figure supplement 1—source data 2.** Original data corresponding to panel C.

**Figure supplement 1—source data 3.** Original data corresponding to panel D.

*2017*; *Zhou et al., 2009*; *Sachs and Sivaselvan, 2015*). PEG 300 effectively reduced ND MCF10DCIS. com cell volume in a dose-dependent manner, showing a greater reduction with 148.8 mOsmol/kg PEG 300 than with 74.4 mOsmol/kg PEG 300 (*Figure 2D*). To assess the impact of PEG-induced cell volume reduction on cell invasiveness, we used a collagen-crosslinked polyacrylamide hydrogel matrix-based invasion assay and exposed cells to 74.4 mOsmol/kg PEG 300 for 2 d. While treatment with 148.8 mOsmol/kg PEG 300 significantly reduced cell viability, cells treated with 74.4 mOsmol/

kg PEG 300 remained viable for 2 d. Monitoring changes in cell invasiveness revealed that exposure of ND MCF10DCIS.com cells to 74.4 mOsmol/kg PEG 300 increased cell invasiveness (*Figure 2E*), confirming the causal relationship between cell volume reduction and increased cell invasiveness.

## Cell crowding induces TRPV4 relocation to plasma membrane in MCF10DCIS.com cells

Cell volume regulation typically depends on osmotic gradients that direct the movement of water across cell membranes (*Jiang and Sun, 2013*). This process is facilitated by ion flux modulation, which is controlled by ion channels and ion transporters located on the plasma membrane (*Jentsch, 2016*; *Matthews et al., 2007*; *Wiggins and Phillips, 2005*; *Martinac and Poole, 2018*; *Zhang et al., 2015*). Notably, cells capable of achieving minimal cell volumes have been shown to successfully invade neighboring tissues (*Watkins and Sontheimer, 2011*). Based on this, we speculated that the high cell volume plasticity observed in MCF10DCIS.com cells underlies their ability to penetrate the surrounding tissues. To achieve such efficient cell volume changes, cell crowding-induced mechanotransduction in MCF10DCIS.com cells may involve modulation of the number of ion channels or transporters on the plasma membrane.

To compare the plasma membrane-associated proteins between ND and OC conditions, we employed mass spectrometry and profiled those proteins using streptavidin-pulled surface-biotinylated cell lysates. *Figure 3A* shows the relative protein densities on the plasma membrane under ND (blue bars) and OC (red bars) conditions in MCF10DCIS.com cells. The figure highlights the top 25 proteins that exhibit over a fivefold increase in expression on the plasma membrane in the OC condition compared to the ND condition (triangle plots). The gene names of these 25 proteins are presented in *Figure 3A*; *Figure 3—source data 7* provides the corresponding protein names along with the corresponding fold increases. Notably, TRPV4, a member of the transient receptor potential family of ion channels known for its mechanosensitive properties (*Ranade et al., 2015*; *Michalick and Kuebler, 2020*), showed a remarkable 153-fold increase in plasma membrane association in response to cell crowding (*Figure 3A*). Additionally, we observed an increase in plasma membrane association of other ion channels, such as SCN11A (the alpha subunit of the voltage-gated sodium channel Nav1.9) with ~42 fold enrichment, and KCNN4 (the small-conductance calcium-activated potassium channel 3 SK3) showing ~ sixfold increase (*Figure 3A*). To evaluate the relative plasma membrane association of proteins under OC versus ND conditions, we performed a comparable mass spectrometry analysis of cell-surface biotinylated cell lysates for MCF10A, MCF10AT1, and MCF10CA1a cells. The top 25 gene/protein names showing increased plasma membrane associations in these cells are listed in *Figure 3—source data 8*. Unlike in MCF10DCIS.com cases, there was no cell-crowding-induced relocation of ion channels and ion transporters to the plasma membrane in MCF10A and MCF10AT1 cells. However, in MCF10CA1a cells, ion transporters such as ATP2B4 showed a ~37 fold greater plasma membrane association under crowding conditions (*Figure 3—source data 8*). This suggests that plasma membrane relocations of these ion channels and ion transporters in response to cell crowding selectively occurred in MCF10DCIS.com cells and, to an extent, in invasive ductal cancer MCF10CA1a cells (*Figure 3—source data 8*).

TRPV4 plays a pivotal role in facilitating the passage of calcium ions (Ca2+) and is integral in detecting various forms of mechanical stresses, including temperature fluctuations and osmotic pressure (*Rosenbaum et al., 2020*; *Lee et al., 2019b*). Activation of TRPV4 channels elevates intracellular calcium levels, leading to cell volume increase via osmotic water influx (*Rosenbaum et al., 2020*; *Becker et al., 2005*). Nav1.9 primarily contributes to the generation and propagation of action potentials in sensory neurons and is associated with various pain disorders (*Dib-Hajj et al., 2015*; *Bennett et al., 2019*; *Huang et al., 2019*). SK3 channels are activated by elevated intracellular calcium levels and serve diverse physiological functions (*Gu et al., 2018*; *Stocker, 2004*). However, unlike TRPV4, Nav1.9 and SK3 channels are not generally classified as mechanosensors. Consequently, we focused on TRPV4, the mechanosensor channel displaying the most significant increase in the plasma membrane association under cell crowding conditions, to further investigate its role in promoting pro-invasive cell volume reduction in MCF10DCIS.com cells.

To confirm the selective plasma membrane association of TRPV4 in OC-treated MCF10DCIS.com cells suggested by the mass spectrometry results, we performed immunoprecipitation using avidin beads on surface-biotinylated lysates from all four cell types under both control and OC conditions,

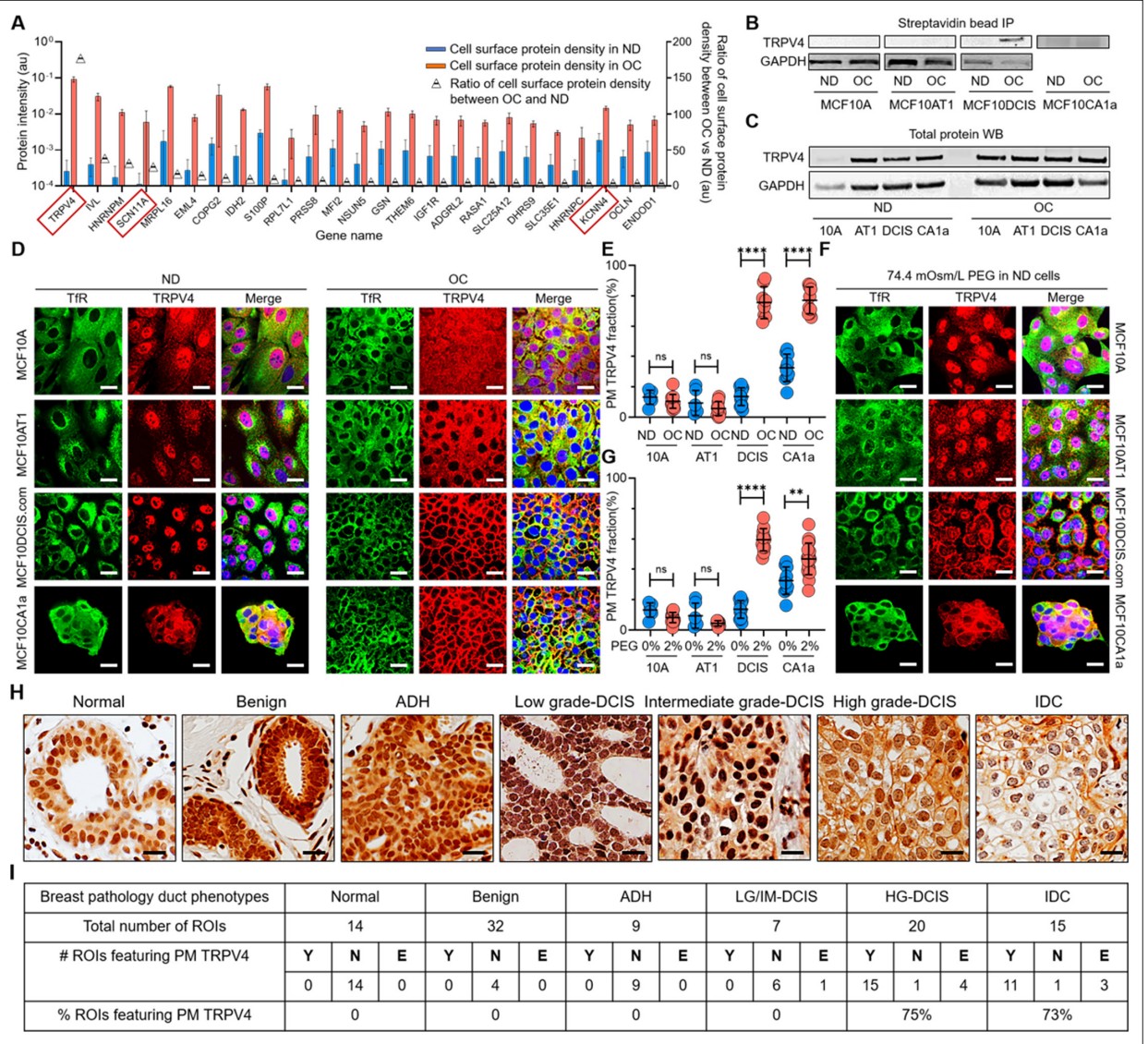

**Figure 3.** Cell crowding induces TRPV4 relocation to plasma membrane in MCF10DCIS.com cells. (**A**) Mass spectrometry data showing proteins enriched in the plasma membrane (PM) > fivefold (fold changes represented using triangle plots; OC/ND ratio on the right axis) when cells are under overconfluence (OC) (red bars) relative to normal density (ND) conditions (blue bars). Ion channels are marked with red boxes, where TRPV4 shows about a 160-fold increased association with the plasma membrane under OC conditions. (**B**) Proteins near and on the plasma membrane were pulled down after cell surface biotinylation with streptavidin beads and immunoblotted for TRPV4. TRPV4 is significantly associated with the plasma membrane in OC MCF10DCIS.com cells. In MCF10CA1a cells, TRPV4 appears to be associated with the plasma membrane under both ND and OC conditions, with a slight increase under OC conditions. (**C**) Immunoblots of whole-protein lysates demonstrate similar overall TRPV4 protein levels across MCF10A cell derivatives, regardless of cell density. This indicates that the differing plasma membrane association of TRPV4 is due to trafficking changes, not expression level changes. GAPDH is used as a loading control. (**D**) Representative immunofluorescence (IF) images by confocal microscopy show TRPV4 (red) localization compared to the control protein transferrin receptor (TfR; green) in MCF10A, MCF10AT1, MCF10DCIS.com, and MCF10CA1a cells under ND and OC conditions. DAPI (blue) staining was used for visualizing the nuclei. As observed in the biochemical data in (**B-C**), cell crowding induces the relocation of TRPV4 to the plasma membrane in MCF10DCIS.com cells. TRPV4 is associated with the plasma membrane in ND MCF10CA1a cells, with a clear elevated association in OC cells. Scale bar = 10 μm. (**E**) Plasma membrane-associated TRPV4 (%) is quantified for the four cell lines under ND and OC conditions by line analysis, showing a significant increase in both MCF10DCIS.com cells and MCF10CA1a cells due to cell crowding. The number of cells used for line analyses (technical replicates merged from three independent experimental repeats) was as follows: MCF10A (ND: 6; OC: 12), MCF10AT1 (ND: 6; OC: 11), MCF10DCIS.com (ND: 12; OC: 8), and MCF10CA1a (ND: 10; OC: 10). (**F**) IF images show that hyperosmotic conditions induced by PEG 300 (74.4 mOsm/Kg) treatment also relocate TRPV4 (red) to the plasma membrane in MCF10DCIS.com cells. TfR localization remains consistent under hyperosmotic conditions. Increased relocation is also observed in MCF10CA1a cells. Scale bar = 10 μm. (**G**) The increased plasma membrane association of TRPV4 due to hyperosmotic stress is quantified by line analysis. The number of cells used for line analyses (technical replicates merged from two independent experimental repeats) was as follows: MCF10A (ND control: 6; ND +4% PEG: 15), MCF10AT1 (ND control:

*Figure 3 continued on next page*

*Figure 3 continued*

6; ND +4% PEG: 9), MCF10DCIS.com (ND control: 12; ND +4% PEG: 8), and MCF10CA1a (ND control: 10; ND +4% PEG: 21). Scale bar = 10 μm. (**H**) Representative regions of interest (ROIs) of TRPV4-stained immunohistochemistry (IHC) images in different pathology phenotypes. High-grade ductal carcinoma in situ (DCIS) and invasive ductal cancer (IDC) ROIs clearly exhibit plasma membrane association of TRPV4. Two high-grade DCIS IHC images were acquired by two different people and both show plasma membrane-associated TRPV4. Scale bar = 20 μm. (**I**) Statistical results from independent histological evaluations of pathologies and TRPV4 distributions of 97 ROIs from 39 patient specimens indicate a high correlation (>70%) of plasma membrane association of TRPV4 with high-grade DCIS or IDC pathologies. Y/N: Yes/no, indicating both pathologists agreed that PM ion channels were present/absent. E: Equivocal, indicating the pathologists disagreed. Significantly high proportions of high-grade DCIS (75%) and IDC (73%) ROIs exhibited plasma membrane TRPV4 association, which was not observed in lower-risk cases. ****$p<0.0001$, ***$p<0.001$, **$p<0.01$, *$p<0.05$, ns: $p>0.05$.

The online version of this article includes the following source data and figure supplement(s) for figure 3:

**Source data 1.** Original data corresponding to panel A.

**Source data 2.** Original data corresponding to panel B.

**Source data 3.** Original data corresponding to panel B.

**Source data 4.** Original data corresponding to panel C.

**Source data 5.** Original data corresponding to panel C.

**Source data 6.** Original data corresponding to panels E and G.

**Source data 7.** Gene and protein names that showed more than a fivefold increase in plasma membrane association under overconfluence (OC) conditions relative to normal density (ND) conditions in MCF10DCIS.com cells were identified by mass spectrometry.

**Source data 8.** Top 25 genes and corresponding proteins that exhibited more than a 100-fold increase in plasma membrane association under OC conditions compared to ND conditions in MCF10A cells (left), and more than a fivefold increase in MCF10AT1 (middle) and MCF10CA1a (right) cells, as identified by mass spectrometry.

**Figure supplement 1.** Binding specificity of TRPV4 antibody.

**Figure supplement 1—source data 1.** Original data for panel B.

**Figure supplement 1—source data 2.** Raw gel image for the immunoblot data.

**Figure supplement 2.** Various ion channels are relocated to the plasma membrane under cell crowding conditions.

**Figure supplement 2—source data 1.** Original data corresponding to panel A.

**Figure supplement 2—source data 2.** Original data corresponding to panel C.

**Figure supplement 2—source data 3.** Original data corresponding to panel C.

**Figure supplement 3.** Plots of the relative intracellular TRPV4 associations between normal density (ND) and overconfluence (OC) or hyperosmotic conditions in all four cell types.

**Figure supplement 3—source data 1.** Original data corresponding to panel A and B.

**Figure supplement 4.** Validation of mechanosensitive relocation of TRPV4 to the plasma membrane in MCF10DCIS.com cells.

**Figure supplement 4—source data 1.** Original data corresponding to panel B.

**Figure supplement 5.** Reversibility of cell-crowding-driven mechanosensing effects in MCF10DCIS.com cells.

**Figure supplement 5—source data 1.** Original data corresponding to panel C.

**Figure supplement 6.** Pathology-dependent differential TRPV4 distributions in patients' immunohistochemistry (IHC) images.

**Figure supplement 6—source data 1.** ROI Images.

followed by immunoblotting. The immunoblots showed a notable association of TRPV4 with the plasma membrane in OC MCF10DCIS.com cells, which was significantly lower in ND cells (*Figure 3B*, *Figure 3—source data 1*). This association was not observed in MCF10A and MCF10AT1 cells under any conditions (*Figure 3B*). Interestingly, in MCF10CA1a cells, TRPV4 associated with the plasma membrane under both conditions, with a slight increase in OC conditions (*Figure 3B*). This increase suggests that these cells may also possess mechanosensing abilities, enabling them to respond to cell crowding. Additionally, immunoblots from whole-cell lysates showed consistent TRPV4 expression levels across different cell types and densities (*Figure 3C*, *Figure 3—source data 2*), indicating that cell crowding influences ion channel trafficking without altering overall expression. This evidence strongly suggests that the relocation of TRPV4 to the plasma membrane is a mechanosensitive response to cell crowding.

To investigate the redistribution of TRPV4 in response to cell crowding, we performed IF imaging with confocal microscopy across all four cell types under both ND and OC conditions. The binding specificity of the TRPV4 antibody was validated in MCF10DCIS.com cells using TRPV4-specific shRNA,

as demonstrated by IF imaging (*Figure 3—figure supplement 1A*) and immunoblotting (*Figure 3—figure supplement 1B*). As expected from the mass spectrometry results, TRPV4 (red) showed only modest localization to the plasma membrane in ND MCF10DCIS.com cells (*Figure 3D*). In contrast, it was prominently associated with the plasma membrane in OC MCF10DCIS.com cells (*Figure 3D*). In MCF10CA1a cells, TRPV4 was also associated with plasma membrane under ND conditions, but cell crowding also increased the association significantly as shown in MCF10DCIS.com cells (*Figure 3D*). Plasma membrane associations were verified using the plasma membrane marker DiIC18(3), which we previously used to visualize the plasma membrane (*Chung et al., 2016*). IF images showed that TRPV4 successfully colocalized with DiIC18(3) in OC MCF10DCIS.com and OC MCF10CA1a cells (*Figure 3—figure supplement 2A*). However, the non-interacting protein transferrin receptor (TfR; green) remained similarly distributed in the OC cells compared to the ND cells, indicating that the protein trafficking changes induced by cell crowding are specific to TRPV4 (*Figure 3D*).

To investigate whether other ion channels identified through mass spectrometry and the well-known mechanosensory channel PIEZO1 (*Syeda et al., 2016*; *Coste et al., 2010*) exhibit similar behavior, we conducted IF imaging on ND and OC MCF10DCIS.com cells targeting KCNN4 and PIEZO1. Unfortunately, due to the absence of specific antibodies, we were unable to examine SCN11. In ND MCF10DCIS.com cells, KCNN4 was observed to be mostly cytosolic, while PIEZO1 was mildly but slightly more associated with the plasma membrane than TRPV4 under ND conditions. However, under OC conditions, both PIEZO1 and KCNN4 also relocated to the plasma membrane, albeit to a lesser extent than TRPV4 (*Figure 3—figure supplement 2B*). The varying degrees of plasma membrane association of PIEZO1 and KCNN4 are quantified using line analysis, as summarized in *Figure 3—figure supplement 2C*.

As expected, in MCF10A, MCF10AT1, and MCF10DCIS.com cells under ND conditions, TRPV4 was modestly present at the plasma membrane but predominantly localized intracellularly, with notable enrichment in the nuclei (*Figure 3D*). Under OC conditions, TRPV4 remained largely cytosolic in MCF10A and MCF10AT1 cells but migrated out of the nuclei (*Figure 3D*). In contrast, in OC MCF10DCIS.com cells, approximately 80% of TRPV4 relocated to the plasma membrane, as shown by line analysis (*Figure 3E*). In MCF10CA1a cells, TRPV4 association with the plasma membrane was evident under both ND and OC conditions, with a further increase under OC, aligning with the immunoprecipitation results (*Figure 3B*). Plasma membrane-associated TRPV4 was quantified using line analysis and plotted in *Figure 3E*. Refer to *Figure 3—figure supplement 3A* for plots of the relative TRPV4 associations with the plasma membrane, cytosol, and nucleus between ND and OC conditions in all four cell types. These IF imaging results underscore the mechanosensitive plasma membrane relocation of ion channels in MCF10DCIS.com, and to an extent, in MCF10CA1a cells, contrasting with the observed insensitivity of MCF10A and MCF10AT1 cells to cell crowding.

To assess the relationship between cell crowding-induced cell volume reduction and plasma membrane relocation of TRPV4, we investigated whether hyperosmotic conditions, which reduce cell volume without cell crowding, would also result in TRPV4 relocation. We subjected ND cells to the same PEG 300 condition (74.4 mOsm/kg for 15 min) used in *Figure 2D* to induce cell volume reduction. In ND MCF10DCIS.com cells, such hyperosmotic cell volume reduction prompted significant plasma membrane relocation of TRPV4, a response not observed in MCF10A and MCF10AT1 cells, but observed in MCF10CA1a cells (*Figure 3F*). Quantitative line analysis confirmed these findings by assessing the relative TRPV4 association with the plasma membrane (*Figure 3G*; *Figure 3—figure supplement 3B* for plots of the relative TRPV4 association with the plasma membrane, cytosol, or nucleus). We confirmed that the apparent plasma membrane-associated TRPV4 under the hyperosmotic condition was indeed localized at the plasma membrane by using an extracellular domain antibody against TRPV4 in live MCF10DCIS.com cells, both untreated and treated with 74.4 mOsm/Kg PEG 300. As expected, significant antibody binding was observed only in PEG 300-treated cells, where intracellular TRPV4 had relocated to the plasma membrane unlike in untreated live cells (*Figure 3—figure supplement 4A–B*). This finding underscores the relationship between mechanosensitive cell volume reduction and plasma membrane relocation of TRPV4, highlighting that both cell crowding and hyperosmotic stress lead to the same effect, which is particularly pronounced in MCF10DCIS.com cells.

We investigated whether TRPV4 relocation to the plasma membrane induced by cell crowding is reversible, as suggested by its impact on invasiveness (*Figure 1—figure supplement 3F*). To test this,

previously OC MCF10DCIS.com cells were reseeded under ND conditions. We then assessed TRPV4 localization via IF imaging to determine if most channels returned to the cytoplasm with only a modest localization at the plasma membrane, and could be relocated to the plasma membrane under mechanical stress, such as hyperosmotic conditions, again. Consistent with their initial ND state, reseeded ND MCF10DCIS.com cells displayed intracellular TRPV4 distribution (*Figure 3—figure supplement 5A*). Upon exposure to hyperosmotic stress (74.4 mOsm/Kg PEG 300), TRPV4 was again relocated to the plasma membrane (*Figure 3—figure supplement 5B*). These findings, quantified through line analysis (*Figure 3—figure supplement 5C*), demonstrate that the mechanosensing response of MCF10DCIS.com cells is reversible.

## Patient tissue analysis shows selective plasma membrane association of TRPV4 in high-grade DCIS cells

MCF10DCIS.com cells represent a basal cell model for high-grade DCIS cells driven by HRAS mutation (*Miller et al., 2000*). However, the majority of patient-derived DCIS cells originate from the luminal cell population and lack the HRAS mutation (*Samson et al., 2021*). Moreover, there are limited options for patient-derived DCIS cell line models, and those that are available are not associated with well-defined pathological grades (*Yong et al., 2014*; *Ransom and Sontheimer, 1995*). Thus, to seek generality of our finding regarding the selective association between plasma membrane relocation and high-grade DCIS pathology, we examined 39 breast cancer patient tissue blocks to assess the selectivity in vivo. We designed an early-stage retrospective study by selecting various breast tissue pathologies that range from benign to invasive cancer. Those pathologies include benign (including usual ductal hyperplasia, papilloma, columnar changes), ADH, low- to high-grade DCIS, and invasive ductal carcinoma (IDC). We also incorporated normal regions for comparison. Hematoxylin and eosin (H&E) staining was used to visualize tissue sections from each patient, excluding samples with prior cancer diagnoses or drug treatments. In total, 97 regions of interest (ROIs) from H&E-stained sections of 39 patient tissue blocks were selected. Two pathologists independently assessed TRPV4 distribution patterns at the single-cell level in corresponding ROIs in the immunohistochemistry (IHC) images (*Figure 3H*). A detailed methodology is described in the Methods section and more example images are shown in *Figure 3—figure supplement 6—source data 1*. High-grade DCIS cells, as depicted in *Figure 3H*, clearly demonstrated plasma membrane-associated TRPV4 (*Figure 3H*), a feature absent in lower-grade DCIS cells (intermediate- and low-grade). This distinction sets them apart not only from cells with ADH and benign pathologies but also from lower-grade DCIS cells. As demonstrated by the IF images in *Figure 3D* for MCF10CA1a (invasive cell mimic), IDC cells exhibited notable plasma membrane TRPV4 (*Figure 3H*), suggesting that they possess a similar pro-invasive mechanotransduction capability to high-grade DCIS cells.

We found that the sensitivity and specificity for plasma membrane TRPV4 in high-grade DCIS cells were 0.75±0.19 (15/20) and 0.98±0.03 (61/62), respectively (95% confidence interval) (*Figure 3I*). It is important to note that the specificity calculation excluded IDC cases. Notably, even in less aggressive pathologies, a significant amount of TRPV4 was localized in the nuclei, as shown for MCF10A and MCF10AT1 cells in the IF images in *Figure 3D*. This underscores our interpretation from the in vitro results in *Figure 3A–D* that increased plasma membrane association of TRPV4 in high-grade DCIS cells results from changes in protein localization through trafficking alterations, rather than differences in expression levels. These initial in vivo results clearly demonstrate the selective and specific association of TRPV4 with the plasma membrane in high-grade DCIS cells. Notably, this association is absent in lower-grade DCIS, ADH, benign, and normal cells, thereby confirming our in vitro findings using MCF10A cell derivatives. These results suggest a potentially critical role for TRPV4 in the progression of high-grade DCIS.

## Cell crowding inhibits ion channels and triggers their plasma membrane relocations

A calcium-permeant ion channel like TRPV4 is known to influence intracellular calcium dynamics (*Becker et al., 2005*). While activation of TRPV4 typically elevates calcium levels and this potentially increases cell volume (*Hoffmann et al., 2009*; *Stutzin and Hoffmann, 2006*; *Clapham, 2007*), the impact of its inhibition is less clear, given the multifaceted nature of calcium signaling in cell volume control (*Jentsch, 2016*). This complexity is compounded by compensatory cellular mechanisms and the

involvement of other ion channels in response to altered calcium homeostasis (*Jentsch, 2016*; *Becker et al., 2005*). Considering these factors, we hypothesized that cell crowding might inhibit calcium-permeant ion channels that are constitutively active at the plasma membrane, including TRPV4, which would then lower intracellular calcium levels and subsequently reduce cell volume via osmotic water movement. To test this, we employed the Fluo-4 calcium assay (*Gee et al., 2000*) to compare relative intracellular calcium levels of MCF10DCIS.com cells between ND and confluent (Con) conditions through 4 X confocal microscopy imaging with 488 nm excitation. For this calcium assay, we opted for confluent conditions instead of OC conditions to collect the Fluo-4 signal from confluent monolayers of MCF10DCIS cells, which demonstrates comparable cell-crowding induced TRPV4 relocation to the plasma membrane (*Figure 4—figure supplement 1A*) as observed in the OC condition. This choice was made because more than one layer of cells was occasionally observed in OC conditions, which would yield an overestimation of crowding-dependent intracellular calcium levels. The intracellular Fluo-4 signal under Con conditions was significantly lower than that under ND conditions, as shown in the fluorescence images of ND versus Con cells in *Figure 4A*. Time-dependent Fluo-4 images were acquired over 25 min. For ND cell images, the average Fluo-4 intensity values were calculated from 10 to 15 selected cells (highlighted in the white box in *Figure 4A*). For Con cell images, intensity values were derived from all cells within the entire 50 μm by 50 μm field of view, as depicted in *Figure 4A*. After background subtraction, the intensity values were plotted in *Figure 4B*. The temporal Fluo-4 intensity profiles from both ND and Con cells remained largely constant over the measured 20 min period. However, Con cells exhibited significantly lower Fluo-4 intensity compared to ND cells, indicating reduced intracellular calcium levels (*Figure 4C*). Lower Fluo-4 intensity in Con cells than in ND cells was not due to limited Fluo-4 reagent in Con samples. Within cell clusters, cells experiencing crowding exhibited notably lower Fluo-4 levels compared to those at the periphery (*Figure 4—figure supplement 1B*), suggesting that cell crowding had resulted in a decrease in intracellular calcium levels, likely due to the mechanosensitive inhibition of calcium-permeable channels like TRPV4.

To investigate whether TRPV4 inhibition was involved in decreased intracellular calcium levels and subsequent relocation of TRPV4 to the plasma membrane, we employed TRPV4-specific pharmacological agents to alter TRPV4 activity and modulate calcium concentrations in ND MCF10DCIS.com cells without applying cell crowding conditions. To inhibit TRPV4 activity, we used a TRPV4 inhibitor (GSK2193874; GSK219) (*Cheung et al., 2017*) at 0.2 and 1 nM, which we confirmed to have insignificant effects on cell viability in both ND and OC conditions for 2 d (*Figure 4—figure supplement 2A*). GSK219 treatments immediately caused a modest dip in the Fluo-4 signal, or intracellular calcium level, followed by a recovery, indicating cellular homeostatic activity. *Figure 4D* shows Fluo-4 images taken before and at the signal dip after applying 1 nM GSK219, where the dip in the Fluo-4 signal is not visually apparent. However, the temporal profile of Fluo-4 intensity in *Figure 4E*, which corresponds to the time points marked in *Figure 4D* ($t_1$: baseline and $t_2$: dip), clearly shows the dip at $t_2$, indicated by ΔCa (the vertical dashed line between the dip and the baseline). This modest Fluo-4 dip at $t_2$ represents the inhibition of activity by GSK219 on a small population of constitutively active TRPV4 channels at the plasma membrane under ND conditions.

In Con cells, 1 nM GSK219 caused a smaller dip in Fluo-4 intensity compared to the one observed in ND cells, with no subsequent changes. This is likely due to fewer constitutively active TRPV4 at the plasma membrane in Con cells than in ND cells. *Figure 4F* shows Fluo-4 images at the baseline and the dip, and *Figure 4G* presents the temporal Fluo-4 intensity profile, highlighting the two-time points. Indeed, the dip in Fluo-4 intensity induced by GSK219 was smaller for Con MCF10DCIS.com cells (mean ΔCa ~ –39±9) than for ND cells (mean ΔCa ~ –49±18), as illustrated in *Figure 4H*. The dose-dependent (0.2 nM and 1 nM GSK219, labeled 219 L and 219 H) and cell density-dependent (ND and Con) Fluo-4 signal dips are summarized by the ΔCa values plotted in *Figure 4H*, showing that the most significant TRPV4 inhibition occurred in ND MCF10DCIS.com cells treated with 1 nM GSK219 (219 H). These findings suggest that a portion of TRPV4 channels that are constitutively active at the plasma membrane became inactive in response to cell crowding and that a significant fraction of TRPV4 channels relocated to the plasma membrane under cell crowding remained inactive.

Administering a TRPV4 activator (GSK1016790A; GSK101) (*Sullivan et al., 2012*; *Baratchi et al., 2019*) immediately caused a notably greater spike in Fluo-4 signal in Con cells than in ND cells, evidencing that a large number of TRPV4 channels at the plasma membrane were inactive in Con cells. In Con cells, 0.2 pM GSK101 led to a large Fluo-4 signal spike with ΔCa ~+2348±597, while in ND

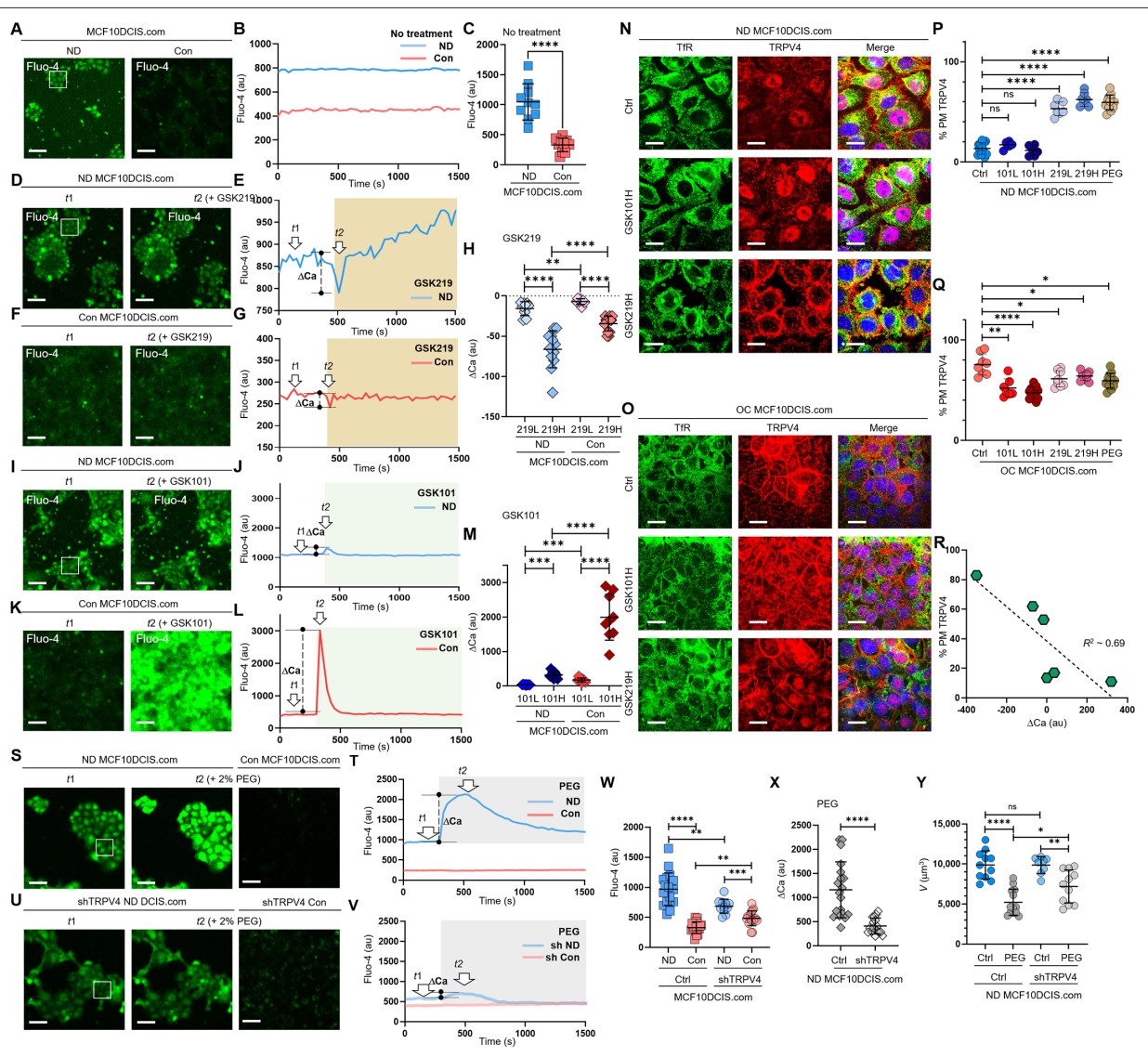

**Figure 4.** Cell crowding inhibits ion channels and triggers their plasma membrane relocations. (**A**) To compare intracellular calcium (Ca²⁺) levels, we used a Fluo-4 AM assay, where green fluorescence intensity increases with higher intracellular Ca²⁺ levels. Calcium levels are significantly lower in confluent (Con) MCF10DCIS.com cells. (**B**) The temporal progression of averaged Fluo-4 intensity in normal density (ND) MCF10DCIS.com cells (blue curve) in the box shown on the left image is compared with that of Con cells (red curve). Fluo-4 intensity is consistently lower in Con cells than in ND cells for approximately 25 min (200 ms acquisition time and 30 s time interval). (**C**) Fluo-4 intensity reduction due to cell crowding is significant in MCF10DCIS.com cells. 10 images were used for calculating average Fluo-4 intensities for both ND and Con cells. (**D-H**) Pharmacological inhibition of TRPV4 with 1 nM GSK219 generates dips in the Fluo-4 signal. Fluo-4 images at the baseline ($t_1$) and the dip ($t_2$) post 1 nM GSK219 are compared in MCF10DCIS.com cells between ND (**D**) and Con (**F**) conditions. The Fluo-4 intensity time traces are compared between ND (blue; E) and Con (red; G) conditions, showing that the magnitude of the dip (marked as ΔCa) is significantly lower in Con cells, where TRPV4 activity is largely inhibited under cell crowding conditions. Notably, the magnitude of ΔCa increased with higher GSK219 doses (1 nM vs. 0.2 nM), but remained significantly lower in Con MCF10DCIS.com cells, with smaller changes observed under the 0.2 nM GSK219 condition. The number of ΔCa measurements (**H**) (technical replicates merged from two independent experimental repeats) was as follows: MCF10DCIS.com (ND+ GSK219 Low: 9; ND+ GSK219 High: 12; Con+ GSK219 High: 10; Con+ GSK219 Low: 14). (**I-M**) TRPV4 activation with 0.2 pM GSK101 leads to a small spike in ND cells (**I, J**). However, in Con cells, the same GSK101 treatment leads to a notably larger spike in Fluo-4 intensity, indicating that TRPV4 inhibition and subsequent relocation to the plasma membrane by cell crowding primes the ion channels for activation. GSK101 treatment also leads to a dose-dependent increase in the spike magnitude with a higher GSK101 concentration being strikingly high in Con MCF10DCIS.com cells (0.05 pM: 101 L; 0.2 pM: 101 H). The number of ΔCa measurements (**M**) (technical replicates merged from two independent experimental repeats) was as follows: MCF10DCIS.com (ND+ GSK101 Low: 7; ND+ GSK101 High: 9; Con+ GSK101 Low: 9; Con+ GSK101 High: 9). (**N-Q**) TRPV4 activation status-dependent intracellular localization changes. (**N**) IF images of TRPV4 (red) and TfR (green) in ND MCF10DCIS.com cells show that GSK101 does not increase plasma membrane association of TRPV4. However, GSK219 significantly relocates TRPV4 to the plasma membrane in a dose-dependent manner (*Figure 4—figure supplement 2B* for all dose

*Figure 4 continued on next page*

*Figure 4 continued*

cases), similar to ND cells treated with 74.4 mOsm/Kg PEG 300. (**O**) In OC cells, while GSK219 does not significantly alter TRPV4 association with the plasma membrane, GSK101 depletes plasma membrane TRPV4 in a dose-dependent manner (***Figure 4—figure supplement 2B*** for all dose cases), suggesting that TRPV4 activation status affects its trafficking. Relative plasma membrane associations with different treatments are quantified for ND (**P**) and OC (**Q**) cells using line analysis. The number of line analyses (P, Q) (technical replicates merged from two independent experimental repeats) was as follows: ND and OC MCF10DCIS.com (control: 12 and 8; GSK219 Low: 6 and 7; GSK219 High: 6 and 13; GSK101 Low: 7 and 8; GSK101 High: 7 and 10; 2% PEG: 12 and 12). (**R**) The values of Fluo-4 spikes by GSK101 and dips by GSK219 show a linear relationship ($R^2 \sim 0.69$) with the plasma membrane TRPV4 association, indicating a negative correlation between them. This reinforces the observation that TRPV4 inhibition increases its association with the plasma membrane, while activation shows the reverse effect. (**S-V**) Compared to the Fluo-4 intensity in control MCF10DCIS.com cells (S, T), shRNA showed similar baseline Fluo-4 levels (**U, V**). However, hyperosmotic stress by 74.4 mOsm/Kg PEG 300 (light gray box) led to a noticeable spike only in control ND cells. Additionally, cell crowding conditions (Con) led to a decreased Fluo-4 level (at $t_1$ baseline in the image in S and red time trace in T); but a reduced in Fluo-4 level difference in shRNA-treated MCF10DCIS.com cells ($t_1$; U) compared to control cases (**S, T**), as shown in the image and time trace ($t_1$; V). (**W**) Relative Fluo-4 time-averaged intensities are plotted for individual control ND (blue) vs. Con (red) cells, and shRNA-treated ND (semi-transparent blue) vs. Con (semi-transparent red) cells. Intracellular calcium levels in shRNA ND cells are lower than those in control ND cells, reflecting the reduced number of TRPV4 channels. The decrease in calcium levels by crowding (Con) in shRNA cells is clearly lower than in control cells, reflecting the importance of TRPV4 in mechanosensing cell volume reduction. The number of Fluo-4 average measurements (technical replicates merged from two independent experimental repeats) was as follows: MCF10DCIS.com control and TRPV4 shRNA groups (ND: 19 and 18; Con: 11 and 17). (**X**) PEG 300-induced calcium spikes are significantly lower in shRNA cells (semi-transparent gray) than in control cells (gray), reinforcing TRPV4's crucial role in MCF10DCIS.com mechanotransduction. The number of ΔCa measurements under 2% PEG 300 condition was as follows: ND MCF10DCIS.com (ND: 19; TRPV4shRNA: 16). (**Y**) TRPV4 silencing significantly reduced the mechanosensing cell volume reduction effect. Control ND cells underwent a 48% volume reduction in response to 74.4 mOsm/Kg PEG 300, whereas TRPV4-silenced cells reduced their volume by only 27%. The number of single-cell volume measurements (technical replicates merged from two independent experimental repeats) was as follows: ND and TRPV4shRNA treated MCF10DCIS.com (Control: 11 and 13; 2% PEG 8 and 11). ****$p<0.0001$, ***$p<0.001$, **$p<0.01$, *$p<0.05$, ns: $p>0.05$.

The online version of this article includes the following source data and figure supplement(s) for figure 4:

**Source data 1.** Original data corresponding to panel B.

**Source data 2.** Original data corresponding to panel C.

**Source data 3.** Original data corresponding to panel E.

**Source data 4.** Original data corresponding to panel G.

**Source data 5.** Original data corresponding to panel H.

**Source data 6.** Original data corresponding to panel J.

**Source data 7.** Original data corresponding to panel L.

**Source data 8.** Original data corresponding to panel M.

**Source data 9.** Original data corresponding to panel P.

**Source data 10.** Original data corresponding to panel Q.

**Source data 11.** Original data corresponding to panel R.

**Source data 12.** Original data corresponding to panel T.

**Source data 13.** Original data corresponding to panel V.

**Source data 14.** Original data corresponding to panel W.

**Source data 15.** Original data corresponding to panel X.

**Source data 16.** Original data corresponding to panel Y.

**Figure supplement 1.** Peripheral cells within MCF10DCIS.com cell clusters exhibit higher calcium levels due to reduced cell crowding effects.

**Figure supplement 1—source data 1.** Original data corresponding to panel A.

**Figure supplement 1—source data 2.** Original data corresponding to panel B.

**Figure supplement 2.** Determination of treatment concentration ranges for TRPV4 activator (GSK101) and inhibitor (GSK219).

**Figure supplement 2—source data 1.** Original data corresponding to panel A.

**Figure supplement 3.** Hyperosmotic stress also induces plasma membrane relocation of ion channels, similar to cell crowding.

**Figure supplement 3—source data 1.** Original data corresponding to panel A.

**Figure supplement 3—source data 2.** Original data corresponding to panel B.

cells, ΔCa was just ~+324±130. This difference is depicted in the Fluo-4 images (*Figure 4I* for ND and *Figure 4K* for Con) and the time-dependent Fluo-4 intensity plots (*Figure 4J* for ND and *Figure 4L* for Con). The dose (101 L and 101 H representing 0.05 pM and 0.2 pM GSK101) and cell-density (ND and Con) dependent Fluo-4 signal spikes are summarized in the plotted ΔCa values (*Figure 4M*). The results show the significant GSK101 dose-responsive activation of previously inactive TRPV4 in Con MCF10DCIS.com cells, suggesting that many inactive TRPV4 channels present at the plasma membrane were primed for activation.

Based on the differing effects of GSK219 and GSK101 on immediate intracellular calcium responses, along with lower baseline calcium levels in Con cells compared to ND cells, we hypothesized that cell crowding induces TRPV4 inhibition, leading to reduced intracellular calcium levels, which in turn triggers the relocation of TRPV4 to the plasma membrane as a compensatory mechanism. To test this hypothesis, we examined TRPV4 localization changes in response to GSK219 or GSK101 in a dose-dependent manner in both ND and OC MCF10DCIS.com cells using IF imaging (*Figure 4N* for higher GSK101/219 concentrations; *Figure 4—figure supplement 2B* for both concentrations).

Indeed, treating ND cells with the TRPV4 inhibitor GSK219 for 1 hr resulted in significant dose-dependent TRPV4 relocation to the plasma membrane (~53±7% at 0.2 nM and ~62±7% at 1 nM), compared to control ND cells (~15±6%), as shown by line analysis (*Figure 4P*). These results indicate that TRPV4 inhibition induces its strategic relocation to the plasma membrane, priming the channels for rapid activation to counteract the effects of mechanically-induced inhibition.

In contrast, in Con cells, where a greater number of inactive TRPV4 channels are likely present at the plasma membrane, GSK101 treatment led to a dose-dependent reduction in plasma membrane-associated TRPV4, likely due to internalization of the activated channels (*Figure 4O and Q*). This finding is consistent with previous findings (*Baratchi et al., 2019*). These data suggest that plasma membrane TRPV4 levels were largely regulated by the channel's activity status. Specifically, channel activation led to the internalization of TRPV4, while channel inhibition promoted the relocation of TRPV4 to the plasma membrane. This relationship was approximately linear, as shown in *Figure 4R*, with intracellular calcium changes inversely proportional to the plasma membrane fraction of TRPV4 ($R^2 \approx 0.69$).

Contrary to our expectation, GSK219 treatment in Con cells slightly reduced TRPV4 levels at the plasma membrane (*Figure 4O and Q*). Moreover, higher doses of GSK219 increased nuclear TRPV4 levels in Con cells without affecting cytoplasmic levels (*Figure 4—figure supplement 2*). These findings suggest that further TRPV4 inhibition under crowding conditions triggers a distinct trafficking alteration. Recent studies have implicated nuclear TRPV4 in regulating nuclear $Ca^{2+}$ homeostasis and $Ca^{2+}$-regulated transcription (*Espadas-Álvarez et al., 2021*), and, in light of this study and our findings, TRPV4 may relocate to the nucleus as a compensatory mechanism to maintain nuclear calcium regulation. This relocation could reflect an adaptive response to preserve calcium-dependent transcriptional programs or other nuclear processes essential for cell survival under mechanical stress.

Given the role of TRPV4 in osmoregulation (*Lee et al., 2019b*; *Liedtke and Friedman, 2003*; *Becker et al., 2005*), we investigated its significant relocation to the plasma membrane under hyperosmotic conditions. In ND MCF10DCIS.com cells, TRPV4 plasma membrane association increased from a baseline of ~15% to ~55% following 74.4 mOsm/kg PEG 300 treatment, as shown by line analysis (*Figures 3E and 4P*). To explore whether TRPV4 inhibition contributed to this relocation, we conducted Fluo-4 assays under the same treatment conditions. The results revealed that osmotic water outflow induced an initial spike in intracellular calcium concentration, followed by a homeostatic relaxation aimed at restoring calcium levels (*Figure 4—figure supplement 3A*). This relaxation likely involved the inhibition of ion channels, including TRPV4, and their subsequent relocation to the plasma membrane.

Consistent with this hypothesis, we confirmed the plasma membrane relocation of other ion channels, namely KCNN4 and PIEZO1, following the same PEG 300 treatment (*Figure 4—figure supplement 3B*, showing IF images and line analysis results). These findings suggest that the increased ion channel association with the plasma membrane under hyperosmotic stress reflects an adaptive response to water outflow, driven by the inhibition and redistribution of channels to regulate osmotic balance.

To investigate the role of TRPV4 in mechanotransduction, we used shRNA to silence TRPV4 gene expression, resulting in a ~50% reduction in TRPV4 protein levels (*Figure 3—figure supplement 1B*).

We then assessed the effects of TRPV4 silencing on intracellular calcium levels in ND and Con conditions, as well as the immediate calcium response to hyperosmotic stress induced by 74.4 mOsm/Kg PEG 300. In ND cells, hyperosmotic stress triggered a significant calcium spike, as shown in the Fluo-4 image (middle; *Figure 4S*) and time trace (blue; *Figure 4T*). In contrast, TRPV4 silencing in ND cells led to a lower intracellular calcium level (left; *Figure 4U*) and a reduced calcium spike in response to hyperosmotic stress (middle; *Figure 4U*, and semi-transparent blue; *Figure 4V*). Furthermore, the effect of TRPV4 silencing on intracellular calcium levels was less pronounced in Con cells (semi-transparent red; *Figure 4V*) compared to the ND counterpart (semi-transparent blue; *Figure 4V*). The relative intracellular calcium levels are summarized in *Figure 4W*.

Notably, TRPV4 silencing significantly reduced the immediate calcium spike as summarized in *Figure 4X*. This calcium spike was attributed to increased calcium concentration due to hyperosmotically reduced cell volume (*Figure 4—figure supplement 3A*). Thus, TRPV4 may play a critical role in mechanotransduction, enabling cells to sufficiently reduce their volume in response to mechanical stress (large volume plasticity). To further investigate this, we examined the effect of TRPV4 silencing on ND MCF10DCIS.com cell volume reduction in response to hyperosmotic stress. We indeed found that control ND cells underwent a 48% volume reduction in response to 74.4 mOsm/Kg PEG 300, whereas TRPV4-silenced cells only reduced their volume by 27% (*Figure 4Y*). Notably, the reduced cell volume change observed in TRPV4-silenced cells under hyperosmotic conditions supports the idea that the observed calcium spikes are driven by cell volume reduction. This finding reinforces the concept that TRPV4 enhances cellular mechanosensitivity through channel inactivation–induced calcium reduction in response to mechanical stress.

## TRPV4 inhibition drives cell volume reduction and increased invasiveness under cell crowding

To elucidate the relationship between TRPV4 inhibition, cell volume reduction, and increased invasiveness under cell crowding, we investigated the effects of TRPV4 activity status on these processes. Specifically, we evaluated how 2 d treatments with GSK101 and GSK219 impacted the cell volumes of ND and OC MCF10DCIS.com cells. This treatment duration was selected to align with the timeframe in which cell volume reduction and invasiveness changes are observed in response to cell crowding.

Under ND conditions, activating TRPV4 with GSK101 (0.05 or 0.2 pM for 2 d), which led to only modest calcium spikes and largely unaltered TRPV4 distribution (*Figure 4I and N*; *Figure 4—figure supplement 2B*), did not significantly impact cell volume (*Figure 5A*). Conversely, TRPV4 inhibition by GSK219 treatment (1 nM for 2 d) that induced greater calcium dips and a significant plasma membrane relocation of TRPV4 (*Figure 4D and N*: *Figure 4—figure supplement 2B*), led to a noticeable cell volume reduction (*Figure 5A*). 0.2 nM GSK219 treatment had a negligible impact on ND cell volume (*Figure 5A*). The GSK219 effect in ND MCF10DCIS.com cell volume was similar to that observed under hyperosmotic conditions by 74.7 mOsm/Kg PEG 300 (*Figure 5A*).

Under Con conditions, GSK101 triggered significant dose-dependent calcium spikes, leading to corresponding increases in cell volume (*Figure 5B*). In contrast, GSK219 under Con conditions had minimal effects on intracellular calcium levels and did not alter TRPV4 distribution at the plasma membrane, resulting in a slight cell volume reduction at 1 nM (*Figure 5B*). Similarly, PEG 300 caused a modest decrease in cell volume, comparable to the effect of 1 nM GSK219 (*Figure 5B*). As expected, plasma membrane-associated TRPV4 (% PM TRPV4) showed a significant inverse linear correlation with cell volume ($R^2 \approx 0.59$) (*Figure 5C*). These findings (*Figure 5A–B*) indicate that increased TRPV4 inhibition and the subsequent reduction in intracellular calcium levels in response to cell crowding are associated with decreased cell volume in MCF10DCIS.com cells.

We investigated the connections between TRPV4 activity, plasma membrane association, cell volume changes, and cell invasiveness. Using a collagen-crosslinked polyacrylamide hydrogel matrix-based invasion assay, we evaluated changes in invasiveness after 2 d treatments with GSK101 or GSK219. In ND MCF10DCIS.com cells, 0.2 pM GSK101 mildly suppressed cell invasiveness, while 0.05 pM GSK101 caused no effect (*Figure 5D*). In contrast, GSK219 increased the invasive cell fraction in a dose-dependent manner under ND conditions, mirroring changes in Fluo-4 signals and plasma membrane-associated TRPV4 (*Figure 5D*). Under OC conditions, GSK101 increased cell volume and suppressed cell invasiveness dose-dependently (*Figure 5E*). Conversely, GSK219 and PEG 300, which slightly reduced cell volume, increased cell invasiveness (*Figure 5E*). Representative invasion images

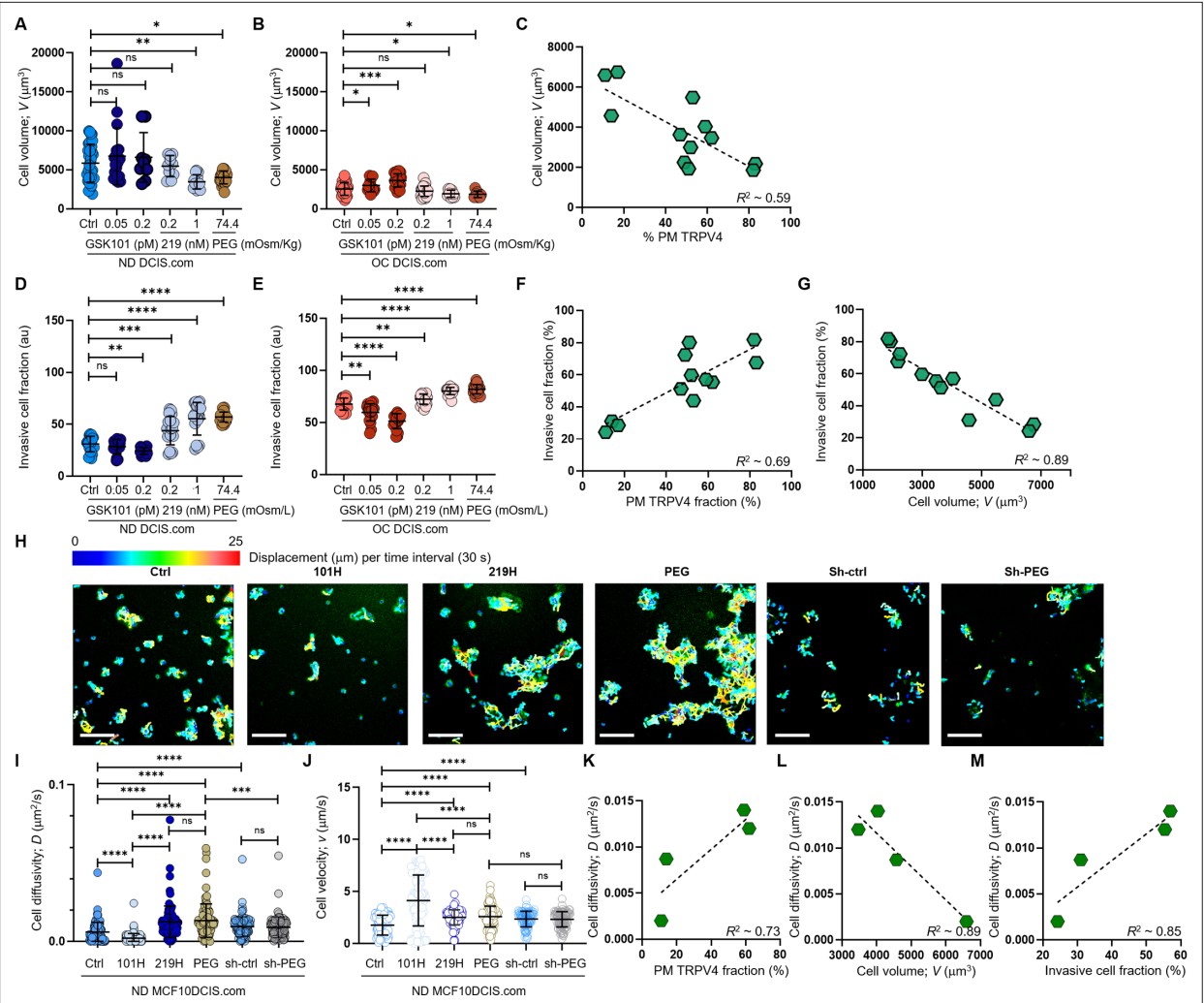

**Figure 5.** Cell crowding-induced plasma membrane TRPV4 association scales with cell volume reduction and increases in invasiveness and motility. (**A-C**). MCF10DCIS.com cell volume changes with TRPV4 inhibition and activation. (**A**) In ND MCF10DCIS.com cells, TRPV4 agonist GSK101, which did not alter plasma membrane association of TRPV4, did not affect cell volume. Conversely, TRPV4 inhibitor GSK219, which increased plasma membrane association in a dose-dependent manner, reduced cell volume, with the effect of 1 nM GSK219 (219 H) being similar to that of 74.4 mOsm/Kg (2%) PEG 300. (**B**) Under OC conditions, GSK101, which led to significant Fluo-4 spikes, increased cell volume in a dose-dependent manner, while GSK219 and PEG only mildly reduced cell volume. (**C**) Cell volume changes in MCF10DCIS.com cells show an inverse relationship ($R^2$=0.59) with plasma membrane association of TRPV4, reflecting the activation status of the channel. The number of single-cell volume measurements (technical replicates merged from three independent experimental repeats): ND (**A**) and OC (**B**) MCF10DCIS.com cells (Control: 33 and 43; GSK101 0.05 pM: 19 and 15; GSK101 0.2 pM: 9 and 22; GSK219 0.1 nM: 10 and 36; GSK101 1 nM: 15 and 9; 2% PEG 300: 23 and 8). (**D-F**) Cell invasiveness increases with greater cell volume reduction and plasma membrane association of TRPV4. (**D**) Cell invasiveness significantly increased with higher GSK219 concentrations under ND conditions. (**E**) GSK101 under OC conditions caused a notable decrease in cell invasiveness in a dose-dependent manner. (**F-G**) Plasma membrane association of TRPV4 predictably reports cell invasiveness ($R^2$~0.69; F), while cell invasiveness and cell volume are inversely related ($R^2$~0.69; G), reinforcing our observation that cell volume reduction promotes cell invasiveness. The number of invasive cell fraction measurements (technical replicates merged from two independent experimental repeats): ND (D) and OC (**E**) MCF10DCIS.com cells (Control: 6 and 4; GSK101 0.05 pM: 4 and 4; GSK101 0.2 pM: 4 and 4; GSK219 0.1 nM: 4 and 4; GSK101 1 nM: 4 and 4; 2% PEG 300: 4 and 7). (**H-M**) To assess if cell motility also follows the trend of cell invasiveness, we performed a single-cell motility assay by tracking nuclear WGA in individual live cells every 60 s for 25 min. (**H**) Representative trajectories of individual cells were color-coded to reflect displacement at each time interval. Compared to untreated ND cells, 0.2 pM GSK101 treatment slowed overall cell diffusion, while 1 nM GSK219 and 74.4 mOsm/Kg PEG 300 treatments increased cell diffusion. ShRNA TRPV4 (Sh-ctrl) increased cell motility under ND conditions. However, with TRPV4 depletion, treatment with 74.4 mOsm/Kg PEG 300 failed to increase cell diffusivity (*D*) in shRNA-treated cells (Sh-PEG), unlike in the untreated cells. Scale bar = 200 μm. Using single-cell analysis, we quantified cell diffusivity (*D*) and speed (*v*; movement directionality). (**I**) GSK101 treatment significantly reduced *D*, while GSK219 and PEG 300 notably increased it. shRNA TRPV4 also increased ND cell *D*, but PEG treatment did not change *D* in the shRNA-treated cells. (**J**) GSK101, GSK219, PEG 300, and shRNA treatments increased v, with GSK101 causing the most significant increase. The directionality of shRNA-treated ND cells was unaffected by PEG treatment. The number of single-cell motility measurements

*Figure 5 continued on next page*

*Figure 5 continued*

of MCF10DCIS.com cells (technical replicates merged from two independent experimental repeats): Control: 81; GSK101: 100; GSK219: 100, PEG: 100; shRNA: 102; shRNA +PEG: 104. (**K**) Like cell invasiveness, cell motility (*D*) positively scales with plasma membrane association of TRPV4 ($R^2$~0.73). (**L**) Cell motility (*D*) inversely relates to cell volume ($R^2$~0.89). (**M**) Cell motility (*D*) and cell invasiveness show a strong linear relationship ($R^2$~0.85), enabling the use of cell motility measurements to assess overall cell invasiveness. ****$p<0.0001$, ***$p<0.001$, **$p<0.01$, *$p<0.05$, ns: $p>0.05$.

The online version of this article includes the following source data and figure supplement(s) for figure 5:

**Source data 1.** Original data corresponding to panel A.

**Source data 2.** Original data corresponding to panel B.

**Source data 3.** Original data corresponding to panel C.

**Source data 4.** Original data corresponding to panel D.

**Source data 5.** Original data corresponding to panel E.

**Source data 6.** Original data corresponding to panel F.

**Source data 7.** Original data corresponding to panel G.

**Source data 8.** Original data corresponding to panel I.

**Source data 9.** Original data corresponding to panel J.

**Source data 10.** Original data corresponding to panel K.

**Source data 11.** Original data corresponding to panel L.

**Source data 12.** Original data corresponding to panel M.

**Figure supplement 1.** Mechanical stresses and TRPV4 activation status affect MCF10DCIS.com cell invasiveness.

are shown in *Figure 5—figure supplement 1A*. These results establish a clear relationship between TRPV4 activity, plasma membrane-associated TRPV4, cell volume, and cell invasiveness. Notably, we observed a significant positive linear correlation between plasma membrane-associated TRPV4 and cell invasiveness (*Figure 5F*; $R^2$~0.69) and an inverse relationship between cell volume and cell invasiveness (*Figure 5G*; $R^2$~0.89).

Next, we investigated cell-crowding induced cell motility changes in MCF10DCIS.com cells, since invasiveness encompasses both the ability to penetrate and migrate through surrounding tissues. As expected from the increased cortical stiffness of OC MCF10DCIS.com cells (*Figure 2C*), we observed that OC MCF10DCIS.com cells underwent cytoskeletal rearrangements, including the increased formation of stress fibers (*Figure 5—figure supplement 1B*), which likely enhanced cell motility (*Tojkander et al., 2012*). To assess how TRPV4 inhibition and activation, which reduced and increased cell volume, respectively, affected the motility of ND MCF10DCIS.com cells and the role of these motility changes on cell invasiveness, we conducted single-cell tracking experiments similar to single molecule tracking methods we previously employed (*Chung et al., 2016*; *Chung, 2017*; *Chung et al., 2010*; *Chung and Mellman, 2014*; *Bien-Ly et al., 2014*). We utilized wheat germ agglutinin (WGA) as a nuclear marker, which stained the cell nucleus within 1 hr of incubation with cells. We then analyzed the diffusivity (*D*) and directionality (*v*) of single-cell movements. We tracked overall single-cell trajectories for 3 hr with 1 min intervals, marking cell displacement over each interval in different colors, ranging from navy for the slowest to red for the fastest (*Figure 5H*). Consistent with the trend in cell invasiveness, TRPV4 activation with 1 pM GSK101 significantly reduced single-cell movements, while inhibition with 0.2 nM GSK219 and 74.4 mOsm/kg PEG 300 notably increased these movements.

Using the single-cell motility assay, we investigated how TRPV4 enhances cell invasiveness through a mechanotransduction mechanism involving cell-volume reduction. Reducing TRPV4 expression (~50%) via shRNA diminished the cell volume reduction effect of mechanosensing (*Figure 4Y*). We predicted that shRNA would also attenuate the increased cell motility effect of mechanosensing in MCF10DCIS.com cells. Additionally, since reduced TRPV4 in ND cells lowered intracellular calcium level as shown in *Figure 4W*, shRNA effect would follow the effect of GSK219. Indeed, shRNA (Sh-ctrl; *Figure 5H and I*) slightly increased both *D* and *v* of MCF10DCIS.com cell movement, mimicking the effects of GSK219. However, the shRNA disabled the mechanotransduction capability of the cells given that there was no increase in *D* and *v* induced by hyperosmotic stress using 74.4 mOsm/Kg PEG 300 (Sh-PEG; *Figure 5H and I*). This result reinforces the critical role of TRPV4 in increasing cell invasiveness through channel activity inhibition-induced cell volume reduction.

Interestingly, as GSK219 and hyperosmotic conditions promoted increased cell movement (*D*; *Figure 5I*), they also slightly increased cell directionality (*v*; *Figure 5J*). Remarkably, cell movement increased in proportion to the amount of TRPV4 at the plasma membrane (*Figure 5K*; $R^2 \sim 0.73$) and decreased with greater cell volume (*Figure 5L*; $R^2 \sim 0.89$). Cell movement and invasiveness showed a strong linear correlation (*Figure 5M*; $R^2 \sim 0.85$), suggesting that these behaviors are governed by the same underlying mechanisms. Thus, our motility assay can inform cell invasiveness at the single-cell level.

## Mechanosensitive TRPV4 relocation to plasma membrane indicates enhanced invasiveness

We investigated whether the mechanosensitive relocation of TRPV4 to the plasma membrane can reliably predict a cell's ability to undergo pro-invasive mechanosensitive cell volume reduction. To achieve this, we used changes in cell motility as markers for overall shifts in cell invasiveness, given the strong correlation observed between cell motility and invasiveness (*Figure 5M*). Our hypothesis is that cells with such mechanotransduction capability, like MCF10DCIS.com cells, should show increased plasma membrane association of TRPV4 when inhibited with GSK219 or under hyperosmotic conditions, evidenced by increased cell motility (*D*) while demonstrating the opposite effect with GSK101 treatment.

For comparison, we employed three uncharacterized cell types: MDA-MB-231, a triple-negative invasive breast cancer cell line, and two recently developed patient-derived DCIS cell lines, ETCC-006 and ETCC-010 (*Samson et al., 2021*; *Yong et al., 2014*), which have not yet been histologically classified by grade. These cells were compared with positive control groups, including MCF10DCIS.com and MCF10CA1a, which were observed to possess pro-invasive mechanotransduction capability (*Figures 5 and 6A-C*). MCF10AT1 served as a negative control, as it lacks this capability (*Figure 6D–F*).

MCF10CA1a cells showed plasma membrane relocation of TRPV4 in response to TRPV4 inhibition (1 nM GSK219; ND 219 H), PEG 300 (74.4 mOsm/kg; ND PEG300), and OC conditions. In contrast, treatment with 0.2 pM GSK101 did not affect TRPV4 plasma membrane association (*Figure 6A*; *Figure 6—figure supplement 1*), showing IF images for TRPV4 (red) and DAPI (cyan). This response was similar to that observed in MCF10DCIS.com cells. The plasma membrane associations of TRPV4 were quantified by line analysis, showing an increase under mechanical stresses (PEG and OC) and TRPV4 inhibition with GSK219 (*Figure 6B*). As expected from the differences in plasma membrane TRPV4 association, MCF10CA1a cells exhibited increased motility (*D*) in response to TRPV4 inhibition (GSK219) and hyperosmotic stress (PEG) (*Figure 6C*). Additionally, their movement became more directional (*v*) with TRPV4 activation (GSK101) (*Figure 6C*), similar to the observations in MCF10DCIS.com cells (*Figure 5I and J*).

In contrast, the negative control MCF10AT1 cells did not exhibit TRPV4 inhibition-dependent plasma membrane relocation of TRPV4, as shown by IF images (*Figure 6D*; *Figure 6—figure supplement 1*) and line analysis results (*Figure 6E*). In these cells, TRPV4 inhibition (GSK219) decreased diffusivity (*D*), while PEG 300 had no effect. Moreover, movement directionality (*v*) remained unchanged under all treatment conditions (*Figure 6F*).

The three test group cells, MDA-MB-231 (*Figure 6G and H*), ETCC-006 (*Figure 6J and K*), and ETCC-010 (*Figure 6M and N*), failed to relocate TRPV4 to the plasma membrane in response to mechanical stresses (PEG and OC) or TRPV4 inhibition (GSK219) (*Figure 6—figure supplement 1*), as shown by IF images and quantification. This suggests that these cells lack the pro-invasive mechanosensitive cell volume reduction capability observed in MCF10DCIS.com cells. Consistently, none of these cells showed increased motility (*D*) or directionality (*v*) under hyperosmotic conditions or TRPV4 inhibition (*Figure 6I, L and O*).

We plotted these results to show cell diffusivity (*D*) scaling with varying plasma membrane associations of TRPV4, where MCF10DCIS.com (green hexagon) and MCF10CA1a (red hexagon) exhibited a positive scaling (*Figure 6P*). However, the changes in plasma membrane TRPV4 occurred over a narrower range (31–79%) for MCF10CA1a compared to MCF10DCIS.com cells (16–75%), reflecting the larger mechanosensitive cell volume plasticity of MCF10DCIS.com cells. In contrast, MDA-MB-231, ETCC-006, and ETCC-010 cells did not show this scaling (*Figure 6Q*). Like MCF10AT1 cells, plasma membrane TRPV4 associations in these cells remained at similar percentages (*Figure 6Q*).

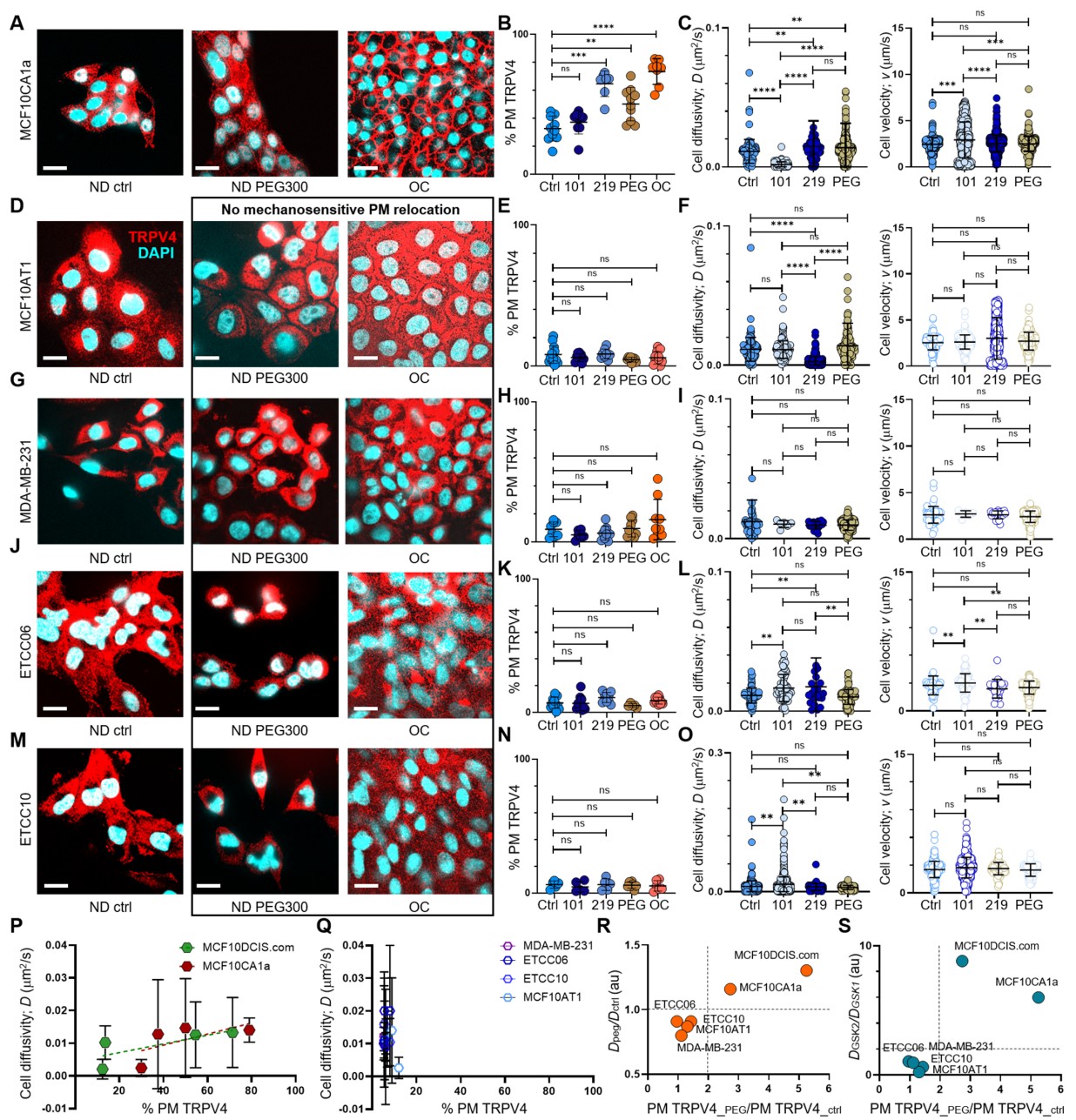

**Figure 6.** pro-invasive cell volume mechanotransduction pathway indicators. (**A**) In MCF10CA1a cells, plasma membrane relocation of TRPV4 was induced by a 15 min treatment with 2% PEG 300 (74.4 mOsm/kg) or by overconfluence (OC) conditions. IF images show a largely intracellular distribution of TRPV4 (red) in normal density (ND) control cells, whereas cells treated with PEG 300 or subjected to OC exhibit a significant increase in plasma membrane-associated TRPV4. Nuclei are stained with DAPI (blue). Scale bars in all panels of this figure represent 10 µm. (**B**) Line analysis quantifying plasma membrane-associated TRPV4 (%) reveals significant increases following a 1 hr treatment with GSK219 (1 nM), a 15 min exposure to 74.4 mOsm/kg PEG 300, or under OC conditions in MCF10CA1a cells. In contrast, no significant increase in plasma membrane TRPV4 is observed with GSK101 treatment. (**C**) In MCF10CA1a cells, cell movement diffusivity (*D*) increased following GSK219 or PEG 300 treatments, whereas it decreased with GSK101. Conversely, movement directionality (*v*) increased significantly with GSK101 (0.2 pM) but remained unchanged with GSK219 or PEG treatments. (**D-O**) No plasma membrane relocation of TRPV4 was observed in response to inhibition by GSK219, hyperosmotic stress induced by PEG 300, or cell crowding (OC) in MCF10AT1 (**D, E, F**), MDA-MB-231 (**G, H, I**), ETCC-006 (**J, K, L**), and ETCC-010 (**M, N, O**) cells. Similarly, GSK219 or PEG 300 did not increase single-cell motility in these cells. This is demonstrated by IF images (TRPV4: red; DAPI: blue) (**D, G, J, M**), line analysis results for plasma membrane-associated TRPV4 (**E, H, K, N**), and single-cell motility analyses for diffusivity (*D*) and directionality (*v*) (**F, I, L, O**). Notably, none of these cell lines showed motility changes in response to PEG 300 treatment. However, responses to TRPV4 activation (GSK101) and inhibition (GSK219) varied across cell types, suggesting distinct roles of TRPV4 in their cancer biology. In MCF10AT1 cells (**F**), GSK219 significantly reduced diffusivity (*D*), while no other treatment

*Figure 6 continued on next page*

*Figure 6 continued*

affected *D* or *v*. In MDA-MB-231 cells (**I**), neither *D* nor *v* was altered by any treatment, indicating that TRPV4 has an insignificant role in their motility. Both ETCC-006 and ETCC-010 cells exhibited increased diffusivity with GSK101; however, GSK219 also increased diffusivity in ETCC-006 cells (**L**), while having no effect on ETCC-010 cells (**O**). Directionality (*v*) increased with GSK101 in ETCC-006 cells (**L**), whereas ETCC-010 cells showed no change in *v* across all conditions (**O**). The number of line analyses for plasma membrane-associated TRPV4 under ND control, ND +0.2 pM GSK101, ND +1 nM GSK219, ND +2% PEG 300, and OC conditions (technical replicates merged from three independent experimental repeats) were: MCF10CA1a (**B**): 11, 6, 8, 12, 10; MCF10AT1 (**E**): 13, 9, 8, 9, 19; MDA-MB-231 (**H**): 13, 9, 8, 9, 19; ETCC-006 (**K**): 12, 10, 10, 5, 10; and ETCC-010 (**N**): 5, 5, 7, 5, 6. The number of single-cell motility analyses under ND control, ND +0.2 pM GSK101, ND +1 nM GSK219, and ND +2% PEG 300 conditions were: MCF10CA1a (**C**): 100, 100, 100, 100; MCF10AT1 (**F**): 130, 161, 582, 183; MDA-MB-231 (**I**): 57, 6, 21, 442; ETCC-006 (**L**): 65, 66, 24, 100; and ETCC-010 (**O**): 317, 1136, 43, 71. (**P**) Plasma membrane association of TRPV4 (% PM TRPV4) scaled positively with cell diffusivity (*D*) over a broader range in MCF10DCIS.com cells compared to MCF10CA1a cells, consistent with the higher cell volume plasticity observed in MCF10DCIS.com cells. This finding suggests that both cell types engage a pro-invasive mechanotransduction pathway. (**Q**) In contrast, this scaling relationship is absent in MCF10AT1, MDA-MB-231, ETCC-006, and ETCC-010 cells, indicating a lack of the mechanotransduction response. (**R**) The presence of this pathway in MCF10CA1a and MCF10DCIS.com cells is further supported by the observed > twofold increase in TRPV4 plasma membrane association (x-axis; PM TRPV4_peg/PM TRPV4_ctrl) and > onefold increase in diffusivity (y-axis; Dpeg/Dctrl) following PEG-300 treatment. (**S**) The cell volume reduction-driven mechanotransduction pathway is further demonstrated by plotting PEG-300-induced changes in TRPV4 plasma membrane association (x-axis; PM TRPV4_peg/PM TRPV4_ctrl) against the diffusivity ratio with GSK219 versus GSK101 (y-axis), where both cell types show a significantly greater than twofold increase, highlighting the activation of this pathway in MCF10DCIS.com and MCF10CA1a cells.****p<0.0001, ***p<0.001, **p<0.01, *p<0.05, ns: p>0.05.

The online version of this article includes the following source data and figure supplement(s) for figure 6:

**Source data 1.** Original data corresponding to panel B.

**Source data 2.** Original data corresponding to panel C.

**Source data 3.** Original data corresponding to panel E.

**Source data 4.** Original data corresponding to panel F.

**Source data 5.** Original data corresponding to panel H.

**Source data 6.** Original data corresponding to panel I.

**Source data 7.** Original data corresponding to panel K.

**Source data 8.** Original data corresponding to panel L.

**Source data 9.** Original data corresponding to panel N.

**Source data 10.** Original data corresponding to panel O.

**Source data 11.** Original data corresponding to panels P and Q.

**Source data 12.** Original data corresponding to panels R and S.

**Figure supplement 1.** Cells lacking the capability for activating pro-invasive mechanotransduction pathway via TRPV4 inhibition-induced cell volume reduction do not relocate TRPV4 to the plasma membrane under TRPV4 inhibition and mechanical stresses.

This dichotomy in the presence of a pro-invasive mechanotransduction program is evident in the plots of diffusivity (*D*) increase versus plasma membrane TRPV4 relocation (*Figure 6R and S*), where only MCF10DCIS.com and MCF10CA1a cells show a significant increase in both parameters. Furthermore, the relationship between plasma membrane-associated TRPV4 and *D* increase with GSK219 versus GSK101 can be used to assess the presence of a pro-invasive, mechanosensitive cell volume reduction program. Most non-mechanotransducing cells remained close to the baseline value of 1, while mechanotransducing cells showed a significant increase in *D* with GSK219 treatment (*Figure 6S*).

## Discussion

Our study explored how cell crowding, a common condition in diseases and development, impacts cell invasiveness. We employed novel assays to quantify cell invasion and single-cell motility in cells with breast tissue pathologies, including DCIS and ADH, which experience cell crowding due to abnormal proliferation in confined ducts. Our results revealed a mechanotransduction pathway activated by crowding selectively in high-grade DCIS cells, establishing a direct link between crowding and invasiveness. This pathway includes ion channel inhibition, intracellular calcium reduction, compensatory channel relocation to the plasma membrane for priming them for later activation, cell volume reduction and cell stiffening, and increased invasiveness and motility (*Figure 7*). Notably, the compensatory relocation of ion channels to the plasma membrane serves as a reliable indicator of an active mechanotransduction pathway. This mechanotransduction capability that enables pro-invasive cell volume

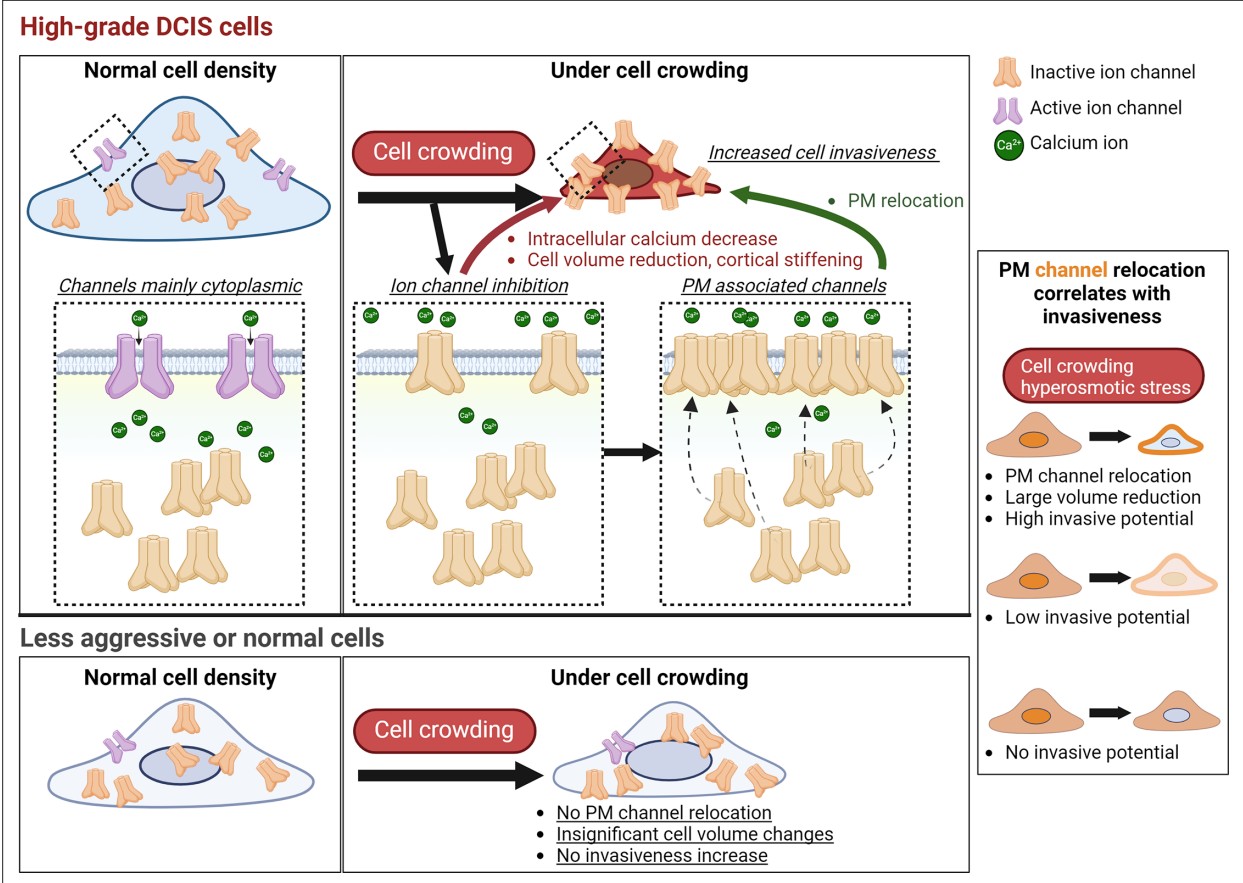

**Figure 7.** Cell crowding activates a pro-invasive mechanotransduction pathway in high-grade ductal carcinoma in situ (DCIS) cells but not in less aggressive or normal cells. This cell-crowding-induced pro-invasive pathway involves a cascade of events, including ion channel inhibition, intracellular calcium reduction, cell volume reduction and cortical stiffening, and increased cell invasiveness and motility. In high-grade DCIS cells, calcium-permeable ion channels such as TRPV4 relocate to the plasma membrane upon inhibition, compensating for reduced intracellular calcium levels by priming the channels for later activation under mechanical stress. The pro-invasive mechanotransduction pathway is selectively triggered by cell crowding or hyperosmotic stress in high-grade DCIS cells, which exhibit significant TRPV4 plasma membrane relocation, pronounced cell volume reduction, and increased motility and invasiveness. In contrast, less aggressive or normal cells remain significantly less or non-responsive to these stimuli. Notably, MCF10DCIS.com cells exhibit greater cell volume reduction compared to other cells, likely due to their larger baseline cell volume at normal density, demonstrating their high cell volume plasticity that correlates with crowding-induced invasiveness. The extent of TRPV4 plasma membrane relocation, cell volume reduction, and increased invasiveness and motility scales with each other, where the increased TRPV4 association with the plasma membrane can robustly serve as a marker of pro-invasive mechanotransduction activation. This mechanotransduction capability sets high-grade DCIS cells apart from less aggressive cells, providing a critical criterion that may help identify high-risk cells with invasive potential. This mechanotransduction capability was also validated in patient specimens, suggesting its relevance in clinical settings. Future investigations will include utilizing TRPV4 localization patterns as a diagnostic tool to assist in pathological grading and as a prognostic marker to identify high-risk DCIS cells likely to activate this pathway under mechanical stress.

reduction distinguishes high-grade DCIS cells from normal, hyperplastic, or lower-grade DCIS cells, supporting previous findings that the ability of cells to reach minimal cell volume promotes efficient invasion (*Watkins and Sontheimer, 2011*).

Using mass spectrometry, we identified TRPV4 as a crucial component of this pathway, providing insights into how non-invasive breast cancer cells may evolve into invasive phenotypes under mechanical stress. Through Fluo-4 assays and TRPV4 activators and inhibitors, we found that cell crowding triggered TRPV4 inhibition, leading to reduced intracellular calcium and subsequent cell volume reduction in MCF10DCIS.com cells. Notably, TRPV4 inhibition prompted its relocation to the plasma membrane, priming it for later activation under suitable signaling conditions to maintain cellular homeostasis. TRPV4 inhibition by cell crowding increased invasiveness and motility, while TRPV4 activation had the opposite effect. Reducing TRPV4 protein expression by 50% via shRNA diminished both calcium decrease and volume reduction under mechanical stress, such as cell crowding and hyperosmotic

conditions. These TRPV4-knockdown cells did not exhibit increased motility under hyperosmotic conditions, demonstrating TRPV4's critical role in mechanotransduction-induced invasiveness through lowering cell volume plasticity. Invasive MCF10CA1a cells showed similar but modest mechanotransduction capability. In contrast, other invasive breast cancer cell lines, such as MDA-MB-231 cells, exhibited opposing responses to TRPV4 inhibition and activation, highlighting the heterogeneity of TRPV4 roles in invasive cancer progression (*Lee et al., 2017*; *Prevarskaya et al., 2010*; *Bai et al., 2023*).

Our discovery of a selective mechanotransduction pathway in high-grade DCIS cells was further validated through the analysis of patient-derived breast cancer tissues. Plasma membrane association of TRPV4 was predominantly observed in most high-grade DCIS and some IDC lesions but was absent in lower-grade DCIS and less aggressive pathologies. This selective relocation highlights TRPV4's critical role in the pro-invasive mechanotransduction pathway unique to high-grade DCIS cells and certain IDC subsets. These findings suggest that plasma membrane-associated TRPV4 may serve as a valuable biomarker for identifying high-grade DCIS lesions.

Our data indicate that ion channels beyond TRPV4 may contribute to pro-invasive mechanotransduction. Notably, SCN11A and KCNN4 showed increased plasma membrane association under cell crowding conditions, suggesting their roles in modulating ion flux and cell volume and thus contributing to the mechanotransduction response. Although PIEZO1 is known to respond to mechanical stimuli (*Syeda et al., 2016*; *Coste et al., 2010*), its lesser plasma membrane relocation compared to TRPV4 under cell crowding and hyperosmotic conditions suggests a lesser role in promoting invasiveness in this case. Further comparisons are needed to determine the dominant ion channels driving mechanotransduction in high-grade DCIS cells. However, it is evident that cell crowding-induced pro-invasive cell volume reduction pathway involves the coordinated action of multiple ion channels to significantly reduce intracellular calcium levels (*White et al., 2016*).

We made the novel discovery that hyperosmotic conditions, which reduce cell volume through osmotic water efflux, mimic the effects of cell crowding by inducing similar pro-invasive cell volume reduction. Treatment with hyperosmotic agents such as PEG 300 caused TRPV4 to relocate to the plasma membrane and increased cell invasiveness, underscoring TRPV4's role as a key mechanosensor responding to external mechanical stresses in this pathway. These findings suggest that the convergence of different mechanical stresses, including cell crowding and hyperosmotic stress, serves as a critical trigger for TRPV4 inactivation–driven, pro-invasive cell volume reduction pathways, activating a unified mechanotransduction signaling cascade.

The study's insights have far-reaching implications, unlocking new research paths in cancer and biological sciences. By elucidating mechanosensitive responses to cell crowding, we can shed light on tissue development, wound healing, and physiological processes involving cell density changes. The parallels between cell crowding and hyperosmotic conditions in driving pro-invasive behaviors warrant further investigation into the intricate mechanisms of cytoskeleton reorganization, ion channel, and transporter relocation, and enhanced cell invasiveness and motility. Importantly, our findings illuminate the mechanotransduction capability of high-grade DCIS cells that differentiate them from less aggressive cells. This understanding paves the way for developing diagnostic and prognostic strategies that leverage the selective intracellular localization patterns of TRPV4 and other mechanosensitive ion channels, ultimately guiding clinical decisions for patients with high-risk DCIS and other cancers.

# Materials and methods

## Key resources table

| Reagent type (species) or resource | Designation | Source or reference | Identifiers | Additional information |
|---|---|---|---|---|
| Cell line (*Homo sapiens*) | MCF10A | ATCC | RRID:CVCL_0598 | Non-tumorigenic human breast epithelial cell line |
| Cell line (*Homo sapiens*) | MCF10AT1 | Karmanos Cancer Institute | RRID:CVCL_5554 | Premalignant variant of MCF10A with oncogenic H-Ras expression |
| Cell line (*Homo sapiens*) | MCF10DCIS.com | Karmanos Cancer Institute | RRID:CVCL_5552 | DCIS progression model with features of comedo necrosis, derived from MCF10AT1 |

*Continued on next page*

*Continued*

| Reagent type (species) or resource | Designation | Source or reference | Identifiers | Additional information |
|---|---|---|---|---|
| Cell line (*Homo sapiens*) | MCF10CA1a | Karmanos Cancer Institute | RRID:CVCL_6675 | Invasive carcinoma derivative of MCF10DCIS.com |
| Cell line (*Homo sapiens*) | ETCC-010 | Leibniz Institute DSMZ | RRID:CVCL_6G22 | Patient-derived, hTERT-immortalized DCIS cell line |
| Cell line (*Homo sapiens*) | ETCC-006 | Leibniz Institute DSMZ | RRID:CVCL_6G19 | Patient-derived, hTERT-immortalized DCIS cell line |
| Cell line (*Homo sapiens*) | MDA-MD-231 | ATCC | RRID:CVCL_0062 | Highly aggressive, triple-negative breast cancer cell line |
| Transfected construct (*Homo sapiens*) | hTRPV4 shRNA Plasmid; shRNA | Santa Cruz | sc-61726-SH | Plasmid construct to transfect and express the shRNA |
| Transfected construct (*Homo sapiens*) | pCSII-EF-miRFP670v1-hGem(1/110) | Vladislav Verkhusha, Addgene | 80006 | Lentiviral construct to transfect and express the gene |
| Antibody | anti-TRPV4 (Rabbit polyclonal) | Abcam | ab39260 | 1:500 (IF), 1:100 (IHC) |
| Antibody | anti-transferrin receptor (Mouse monoclonal) | Thermo Fisher | 13–6800 | 1:500 |
| Antibody | anti-Piezo1 (extracellular) (Rabbit polyclonal) | Alomone | APC-087 | 1:500 |
| Antibody | anti-KCNN4 (K$_{Ca}$3.1, SK4) (extracellular) (Mouse monoclonal) | Alomone | ALM-051 | 1:500 |
| Antibody | anti-Rabbit IgG (H+L) Highly Cross-Absrobed Secadary Antibody, Alexa Fluor Plus 555 (Goat polyclonal); anti-rabbit-Alexa 550 | Thermo Fisher | A32732 | 1:5,000 |
| Antibody | anti-Mouse IgG (H+L) Highly Cross-Absrobed Secadary Antibody, Alexa Fluor 488 (Goat polyclonal); anti-mouse-Alexa 488 | Thermo Fisher | A11029 | 1:5,000 |
| Antibody | anti-TRPV4 (Rabbit polyclonal) | Thermo Fisher | PA5-41066 | 1:1,000 |
| Antibody | anti-GAPDH (6C5) (Mouse monoclonal) | Santa Cruz | sc-32233 | 1:2,000 |
| Antibody | anti-Mouse IgG (H+L) Superclonal Secondary Antibody, HRP (Goat superclonal) | Thermo Fisher | A28177 | 1:4,000 |
| Antibody | anti-Rabbit IgG (H+L) Superclonal Secondary Antibody, HRP (Goat superclonal) | Thermo Fisher | A27036 | 1:4,000 |
| Antibody | Anti-TRPV4 (extracellular) (Rabbit polyclonal) | Alomone | ACC-124 | 1:300 |
| Peptide, recombinant protein | Streptavidin UltraLink Resin; resin | Thermo Fisher | 53114 | High-capacity streptavidin resin for affinity purification of biotinylated proteins |
| Commercial assay or kit | Lenti-X Packaging Single Shots (Ecotropic) | Takara Bio | 631278 | Lentiviral packaging system for efficient viral particle production in ecotropic host cells |
| Commercial assay or kit | BCA Protein Assay Kits; bicinchoninic acid assay | Pierce | 23225 | Quantitative protein assay based on colorimetric detection of peptide bonds |
| Commercial assay or kit | Fluo-4 Direct Calcium Assay Kit | Thermo Fisher | F10471 | Fluorescent assay for real-time calcium flux measurement in live cells |
| Chemical compound, drug | GSK 1016790 A; GSK101 | Tocris | 64T33 | TRPV4 agonist |
| Chemical compound, drug | GSK 2193874; GSK219 | Tocris | 5106 | TRPV4 antagonist |
| Chemical compound, drug | Polyethylene glycol 300; PEG 300 | Millipore Sigma | 8074845000 | Osmotic reagent used to induce hyperosmotic stress |
| Chemical compound, drug | Polybrene Infection/Transfection Reagent; polybrene | Sigma-Aldrich | TR-1003–50 UL | 8 µg/mL |
| Chemical compound, drug | Puromycin Dihydrochloride, puromycin | Thermo Fisher | A1113803 | 1 µg/mL |
| Chemical compound, drug | Poly-L-Lysine | Fisher Scientific | A005C | 50 µg/mL |

*Continued*

| Reagent type (species) or resource | Designation | Source or reference | Identifiers | Additional information |
|---|---|---|---|---|
| Chemical compound, drug | Glutaraldehyde | VWR | 100503–974 | 0.5% |
| Chemical compound, drug | fluorescein-gelatin | Thermo Fisher | G13187 | 1 mg/mL |
| Chemical compound, drug | Gelatin | Sigma-Aldrich | G2500 | 4 mg/mL |
| Chemical compound, drug | EZ-Link(TM) Sulfo-NHS-Biotin | Thermo Fisher | A39256 | 2 mM |
| Other | DAPI and Hoechst Nucleic Acid Stains; DAPI | Thermo Fisher | D3571 | 300 nM, Immunofluorescence, Line analysis, Cell invasion assay, |
| Other | DiI Stain (1,1'-Dioctadecyl-3,3,3',3'-Tetramethylindocarbocyanine Perchlorate ('DiI';DiIC$_{18}$(3))) | Thermo Fisher | D282 | 5 µM, Materials and Methods – Immunofluorescence, Line analysis |
| Other | 8-well chambered slides | Nunc Lab-Tek | 155409PK | Materials and Methods – Live cell detection, Cell invasion assay, Calcium reporter assay, Cell viability assay |
| Other | Q Exactive HF mass spectrometer | Thermo Fisher | | Materials and Methods – Mass spectrometry |
| Other | MatTek 35 mm dishes with No. 1.5 coverslip, 14 mm glass diameter, and Poly-D-Lysine coated glass bottoms | MatTek | P35GC-1.5–14 C | Materials and methods – Cell motility Assay |
| Other | WGA-488 | Thermo Fisher | W11261 | 10 µg/mL (motility assay), 1 µg/mL (viability assay), Materials and methods – Cell motility assay; Cell viability assay |
| Other | Propidium Iodide Ready Flow Reagent; propidium iodide; PI | Thermo Fisher | R37169 | Materials and methods – Cell viability assay |

## Cell lines and treatments

MCF10A cells were purchased from ATCC, and MCF10AT1, MCF10DCIS.com, and MCF10CA1a cells were obtained from the Barbara Ann Karmanos Cancer Institute through a material transfer agreement. ETCC-006 and ETCC-010 cells were purchased from the Leibniz Institute DSMZ, and MDA-MB-231 cells were from ATCC. All cell lines were authenticated prior to purchase, used at a low passage number (<15), and confirmed to be mycoplasma-free by PCR analysis. MCF10A and MCF10AT1 cells were maintained in DMEM/F12 medium supplemented with 5% horse serum, 20 ng/mL EGF, 0.5 µg/mL hydrocortisone, 10 µg/mL insulin, and 1% penicillin and streptomycin. Additionally, MCF10A cells were supplemented with 100 ng/mL cholera toxin. MCF10DCIS.com and MCF10CA1a cells were cultured in DMEM/F12 with 5% horse serum and 1% penicillin and streptomycin. ETCC-006 and ETCC-010 cells were maintained in RPMI 1640 supplemented with 10% fetal bovine serum (FBS) and 1% penicillin and streptomycin. MDA-MB-231 cells were cultured in DMEM/F12 with 10% FBS and 1% penicillin and streptomycin. All cells were cultured at 37 °C in 5% CO$_2$. For TRPV4 modulation, we used the TRPV4 activator GSK1016790A (GSK101; Tocris 64T33) and the TRPV4 inhibitor GSK2193874 (GSK219; Tocris 5106).

For volume measurements, parental cells were mixed with those expressing RFP at a 9:1 ratio to distinguish individual cells. Cells were treated with 2% (74.4 mOsmol/kg) or 4% PEG 300 (148.8 mOsmol/kg; v/v; Millipore Sigma 8074845000) in culture medium for 15 min, 2% PEG- 00 for 48 hr, GSK1016790A (Tocris; 0.05 and 0.2 pM), or GSK2193874 (Tocris; 0.2 and 1 nM) for 48 hr.

## TRPV4 knockdown by shRNA

Cells in growth medium without penicillin and streptomycin were transfected with the hTRPV4 shRNA plasmid (1 and 2 µg; Santa Cruz sc-61726-SH) and Lipofectamine3000 in opti-MEM. At 6 hr after transfection, the medium was replaced, and cells were cultured for 30 hr before processing for western blot and immunofluorescence analyses.

## Immunofluorescence

The cells were fixed with 4% paraformaldehyde (Fisher Scientific) for 20 min, followed by permeabilization with 0.1% saponin (Fisher Scientific) for 5 min at room temperature (RT). After permeabilization,

the cells were washed three times in PBS and blocked in 0.1% saponin +10% BSA in PBS at RT for 1 hr. Cells were incubated with primary antibodies overnight at 4 °C in 0.1% saponin in PBS. The primary antibodies used were TRPV4 (Abcam 39260; different lot numbers were tested and selected for optimal performance; 1:500 dilution), TfR (Thermo Fisher Scientific 13–6800; 1:500 dilution), PIEZO1 (Alomone APC-087; 1:500), and KCNN4 (Alomone ALM-051; 1:500). Cells were washed three times in PBS, incubated with fluorescent-tagged anti-rabbit-Alexa 555 or anti-mouse-Alexa 488 secondary antibodies (Thermo Fisher Scientific) and DAPI for 1 hr at RT, imaged by confocal microscopy. For plasma membrane staining, DiIC18(3) (1,1'-Dioctadecyl-3,3,3',3'-Tetramethylindocarbocyanine Perchlorate; DiIC18(3); Thermo Fisher Scientific, Cat # D282) was diluted in warm PBS to 5 µM. Media was removed from the cells, then the 5 µM DiIC18(3) in PBS was added. The dish was transferred to the incubator (37 C, 5% $CO_2$) for 10 min. Following the 10 min incubation, the DiIC18(3) in PBS was removed and the cells were washed 2 x with complete cell growth media. The cells were then fixed with 2% PFA in PBS.

## Confocal imaging and image processing

2D images were acquired using a Yokogawa spinning-disk confocal microscope (Andor Technology) installed in a Nikon Eclipse TE2000 inverted microscope using a 60 x/1.49NA Plan Apo objective (Nikon). The samples were illuminated using 430, 488, 561, and 647 nm solid-state lasers (Andor Technology). Images were acquired using an iXon back-illuminated EMCCD camera (Andor Technology). For volume measurements, 3D confocal images were acquired using z-step sizes calculated based on Nyquist conditions. Images were processed using ImageJ (NIH) or Imaris (Bitplane) software. For live cell imaging, cells were maintained in 5% $CO_2$ at 37 °C in a stage-top incubator (Oko-lab).

## Western blotting

ND cells (40–60% cell density) were lysed in SDS sample buffer, and OC cells were lysed in cytoskeleton buffer (10 mM Tris pH 7.4,100 mM NaCl, 1 mM EDTA, 1 mM EGTA, 1% Triton X-100, 10% glycerol, 0.1% SDS, 0.5% deoxycholate). SDS sample buffer (Bio-Rad) supplemented with reducing reagent was added to lysates and boiled at 100 °C for 10 min. The samples were separated on 4–15% SDS-PAGE gels and transferred to nitrocellulose membranes (Bio-Rad). Membranes were blocked in TBST (TBS +0.1% Tween-20) containing 5% non-fat milk or 10% BSA and then incubated with primary TRPV4 antibody (Thermo Fisher Scientific PA541066; 1:1000) and GAPDH (Santa Cruz sc-32233; 1:2000) overnight at 4 °C. Membranes were washed three times in TBST (5 min each) and then incubated with horseradish peroxidase (HRP)-conjugated anti-rabbit (Thermo Fisher Scientific A28177) or anti-mouse (Thermo Fisher Scientific A28177) secondary antibodies for 30 min at RT. Blots were visualized using West Femto maximum sensitivity chemiluminescent substrate (Thermo Fisher Scientific).

## Lentiviral infection and stable cell generation

To produce lentiviruses encoding RFP, HEK-293 cells were transfected with the lentiviral vector plasmid DNA using Lenti-X Packaging Single Shots (Takara). pCSII-EF-miRFP670v1-hGem(1/110) was a gift from Vladislav Verkhusha (Addgene plasmid # 80006; http://n2t.net/addgene:80006; RRID:Addgene_80006). Supernatants containing lentiviral pseudoparticles were harvested 24 and 48 hr posttransfection. Harvested lentiviral particles were immediately stored at −80 °C. To establish stable cell lines, 70% confluent cells at 1 d post-seeding were infected with lentivirus in the presence of 8 µg/mL polybrene (Sigma-Aldrich). Two days after transduction, 1µg/mL puromycin (Thermo Fisher Scientific) was added to the medium to select for stably transduced cells. The samples were visualized daily to ensure that the untransduced cells in the wells were not viable. Once the polyclonal populations had sufficiently expanded, cell stocks were prepared and harvested for protein expression assays.

## Cell invasion assay

For cell invasion assays, 8-well chamber slides (Nunc Lab-Tek) were coated with 50 µg/mL poly-L-lysine (Fisher Scientific) for 30 min and washed with PBS. The slides were fixed with 0.5% glutaraldehyde (VWR) for 20 min and washed. Gelatin was conjugated with fluorescein by mixing 200 µL fluorescein-gelatin (1 mg/mL; ThemoFisher Scientific) with 800 µL unlabeled gelatin (4 mg/mL) (Sigma-Aldrich), followed by incubation for 5 min at 60 °C. The gelatine mix was allowed to cool at RT for 5 min and then applied, and the slides were incubated for 15 min. Slides were washed with PBS and disinfected

with 70% ethanol for 30 min. Following three PBS washes, the residual reactive groups were quenched with growth medium by incubating at room temperature for 30 min in the dark. Cells were seeded in a fresh growth medium and incubated undisturbed on a horizontal surface at RT for 30 min to encourage cell distribution. Chamber slides were incubated with 5% $CO_2$ at 37 °C. To detect cell invasion, cells were fixed with 4% formaldehyde and stained with DAPI (Thermo Fisher Scientific), and imaging was performed using a 4 x/0.2 NA Plan Apo objective (Nikon). Cell invasion was determined by quantifying the sites of the degraded matrix, which were visible as dark areas in the bright-green fluorescent gelatin matrix. The area of gelatin digestion and the number of cells were calculated using the ImageJ software.

## Nanoindenter assay for measuring cortical stiffness

Stiffness measurements were performed on live cells in 5% $CO_2$ at 37 °C, maintained in a stage-top environmental chamber (Oko-lab) using a nanoindenter (Optics11 Chiaro) attached to the confocal microscope. Nanoindentation was performed at the cell surface of single cells with an indentation probe with a spring constant (0.24 N/m) and tip diameter (10 μm). The Hertzian contact model was used to fit the data to extract Young's modulus in the elastic regime.

## Surface biotinylation and pull-down assay

ND cells that were 40–70% confluent and OC cells were detached and washed twice with PBS. The cells were resuspended at a concentration of $25 \times 10^6$ cells/mL in PBS containing 2 mM sulfo-NHS-biotin (Invitrogen). The reaction mixture was then incubated at RT for 60 min. Cells were washed three times with 1 M Tris pH 8.0 to quench and remove the excess biotin reagent. Cell pellets were lysed in 1.0 mL of cold lysis buffer (50 mM HEPES pH 7.2, 150 mM NaCl, 1.0% Triton X-100, 1.0% CHAPS, 100 x protease and phosphatase inhibitors). Lysates were incubated on ice for 30 min, transferred to tubes, and centrifuged ($10,000 \times g$ for 5 min) to remove insoluble material. Supernatants were collected, and protein concentrations were measured using the bicinchoninic acid assay (Pierce). We washed 50 μL resin (Thermo Fisher Scientific) with immobilized streptavidin twice with binding buffer (0.1 M phosphate, 0.15 M NaCl pH 7.2) and centrifuged at $5000 \times g$ for 1 min. Biotin-labeled cell lysate (1 mg) was added to the resin and incubated with rotation for 1 hr at RT. The resin was washed by resuspending in binding buffer, centrifuging to pellet the resin, and removing the supernatant by aspiration. The wash was repeated four times. Samples were boiled in SDS-PAGE sample buffer with DTT and separated by electrophoresis.

## Mass spectrometry

Peptide mixtures from each sample were analyzed by LC-MS/MS using a nano-LC system (Easy nLC1000) connected to a Q Exactive HF mass spectrometer (Thermo Fisher Scientific). The platform was configured with a nano-electrospray ion source (Easy-Spray, Thermo Fisher Scientific), an Acclaim PepMap 100 C18 nanoViper trap column (3 μm particle size, 75 μm ID ×20 mm length), and an EASY-Spray C18 analytical column (2 μm particle size, 75 μm ID ×500 mm length). Peptides were eluted at a flow rate of 300 nL/min using linear gradients of 5–25% acetonitrile (in aqueous phase and 0.1% formic acid) for 40 min, followed by 45% for 10 min, and static flow at 90% for 10 min. Mass spectrometry data were collected in a data-dependent manner, switching between one full-scan MS mode (m/z:380–1400; resolution: 60 K; AGC:3e6; maximum ion time: 20ms), and 10 MS/MS scans (resolution: 15 K; AGC:1e5; maximum ion time: 120 ms; nCE:27) of the top 10 target ions. Ions were sequenced once and dynamically excluded from the list for 20 s. The datasets were searched using MaxQuant at default parameters against the UniProt Human Proteome database.

## Line analysis

IF images of ion channels (TRPV4, PIEZO1, or KCNN4), TfR, DiIC18(3), and DAPI were opened in ImageJ. These images were concatenated so that a line crossing a cell in all three images could be used to measure an approximate percentage of protein localization in the plasma membrane, cytosol, or nucleus. Background was subtracted from all images. DiIC18(3) signal was used as a guide for plasma membrane location (between 50% of the peak intensity) and DAPI signal (between 50% of the peak intensity) for nuclear locations. The cytosol location was defined as the region between the 50% peak intensity points of the DiIC18(3) and DAPI signals. The ion channel signal in each window along

the line was then averaged. The percentage of protein localization in the plasma membrane, cytosol, or nucleus for a cell was calculated by dividing the average intensity of each location by the sum of intensities in all three locations.

## Live cell detection of plasma membrane-associated TRPV4

Cells were seeded in a LabTek II 8-well Chambered Coverglass dish (Thermo Fisher 155409) at a density of 10,000 cells per well and allowed to adhere and grow for 2 d. Afterward, the cells were incubated on ice for 1 hr with 1:300 Anti-TRPV4 (extracellular) rabbit primary antibody (Alomone ACC-124) in either complete media or 74 mOsm/Kg PEG 300 in complete media. Following the incubation, the cells were washed twice with the respective media (complete media or 74 mOsm/Kg PEG 300 in complete media) while remaining on ice. Subsequently, the cells were stained for 1 hr at 22 °C with 1:5000 Donkey anti-Rabbit IgG H+L Highly Cross-Adsorbed Secondary Antibody, Alexa Fluor 647 (Thermo Fisher A31573) in the same respective media. After the staining period, the cells were imaged.

## Pipeline for analyses of patient histologic samples

1. Specimen collection: A retrospective clinical study was conducted to collect patient tissue blocks under IRB approval from George Washington University (IRB# NCR203065), with patient consent waived. Once the tissue specimens and associated clinical data for the study are identified, a code will be assigned to de-identify the patient specimens and clinical information for further study and analysis. The code links will be kept in a password-protected file in a secure location, physically separated from the database where the data was collected. Further use and analysis of tissues and clinical data will only be done using the assigned code. A pathologist chose 39 patient breast formalin-fixed paraffin-embedded (FFPE) tissue specimens, representing a combination of the following pathologies: benign, ADH, IDC, low-, intermediate-, or high-grade DCIS, and IDC, along with normal tissue from each patient. Specimens associated with a past cancer diagnosis or drug treatment history were excluded.

2. Sample preparation for histopathologic evaluation: For each FFPE block, two serial dissections were carried out. Tissue sections on unlabeled slides were stained with H&E and subjected to immunohistochemical (IHC) staining using anti-TRPV4 antibodies (Abcam 39260; 1:100 dilution). Whole-slide images were captured using the Olympus VS200 whole-slide imaging system. To validate the binding specificity of the antibody, both negative and positive control samples were prepared. IHC staining was performed by VitroVivo Biotech (Rockville, MD) and Histoserv, Inc (Germantown, MD).

3. Two sequential annotations: A breast surgical pathologist annotated the pathology stages within specified regions of interest (ROIs) on whole slide H&E images, based on cell morphological features. Two pathologists independently evaluated the same ROIs in the corresponding IHC images. They annotated the protein distribution using a three-tier classification:

Case 1: Absence of TRPV4.
Case 2: Intracellular TRPV4 localization.
Case 3: Presence of TRPV4 in the plasma membrane, with or without intracellular TRPV4.

The independent annotations concerning protein localization by the two pathologists were then subjected to statistical tests for selectivity and specificity. Any IHC ROIs with classification disagreements between the pathologists were designated as equivocal cases.

## Calcium reporter assay

This assay was completed using the Fluo-4 Direct Calcium Assay Kit (Thermo Fisher F10471), including the assay buffer (Thermo Fisher), 2 X Fluo-4 calcium assay reagent (Thermo Fisher), and preweighed, water-soluble probenecid (Thermo Fisher P36400; 2.5 mM). The 2 X Fluo-4 calcium assay reagent was first prepared by adding 10 mL of the Fluo-4 calcium assay buffer and 200 µL of 250 mM probenecid to the desiccated calcium assay reagent at room temperature (21 °C). This mixture was then vortexed and allowed to sit for 15 min. While waiting for the 2 X Fluo-4 calcium assay reagent to finish being

prepared, the media from the well of a LabTek II Chambered Coverglass with Cover #1.5 Borosilicate Sterile 8-well plate (Thermo Fisher 155409) was removed and replaced with 200 µL of complete media. Once the 2 X Fluo-4 calcium assay reagent was fully prepared, 200 µL of the reagent was added to the well with 200 µL of complete media. The 8-well plate was then placed at 37 °C and 5% $CO_2$ for 30 min. Following 30 min at 37 °C and 5% $CO_2$, the 8-well plate was removed from the incubator and placed at room temperature for 30 min. After incubation in the reagent for 30 min at room temperature, the 8-well plate was imaged using the confocal microscope with a 488 nm laser for 30 min with 30 s intervals between images and an exposure time of 200 ms. If the sample was treated, the sample was imaged for 35 min with 30 s intervals, with the treatment being added after 5 min had elapsed. Added treatments all had a final volume of 200 µL and were composed of: 1 nM GSK 2 in 1:1 media:2 X Fluo-4 calcium assay reagent, 0.2 nM GSK 2 in 1:1 media:2 X Fluo-4 calcium assay reagent, 0.2 pM GSK 1 in 1:1 media:2 X Fluo-4 calcium assay reagent, 1:1:1 water:media:2 X Fluo-4 calcium assay reagent, and 0.3 Osm/L PEG 300 in 1:1 media:2 X Fluo-4 calcium assay reagent.

## Cell motility assay

Cells were first seeded in MatTek 35 mm dishes with No. 1.5 coverslip, 14 mm glass diameter, and Poly-D-Lysine coated glass bottoms (MatTek P35GC-1.5–14 C). Once the cells adhered, they were stained with 10 µg/mL WGA-488 (Thermo Fisher W11261) for 10 min at room temperature. Following the staining, the cells were washed with complete media twice and then incubated at 37 °C and 5% $CO_2$ for 1 hr. During this incubation, the cells received the following treatments: 0.2 pM GSK 101 in complete media for 1 hr, 0.05 pM GSK 101 in complete media for 1 hr, 1 nM GSK 219 in complete media for 1 hr, 0.2 nM GSK 219 in complete media for 1 hr, 74.4 mOsm/Kg PEG 300 in complete media for 15 min, and 1:1 hypoosmotic solution in complete media for 15 min. After the incubation, the sample was placed in an OkoLab chamber at 37 °C and 5% $CO_2$ for imaging. Imaging was completed using a confocal microscope with a 488 nm laser for 3 hr with 1 min intervals.

## Cell viability assay

MCF10DCIS.com cells were plated in a LabTek II 8-well Chambered Coverglass dish (Thermo Fisher 155409) at a seeding density of 10,000 cells per well. Once the cells in the well reached the desired number of days post-confluence, they were stained using Propidium Iodide Ready Flow Reagent (Thermo Fisher R37169). Specifically, one drop of propidium iodide (PI) was added to 500 µL of cell growth medium per well and incubated for 15 min at room temperature (22 °C). After staining, the cells were washed twice with complete growth medium. Next, the cells were stained with 1 µg/mL WGA-488 (Thermo Fisher w11261) in complete growth medium for 15 min at 37 °C in a 5% $CO_2$ atmosphere. Following the incubation with WGA-488, the cells were washed three times with complete growth medium and then incubated for 2 hr at 37 °C in 5% $CO_2$ to allow WGA migration into the nuclei. Finally, the cells were imaged using confocal microscopy at 4 X magnification, with laser excitation at 488 nm for WGA-488 and 560 nm for propidium iodide.

## Acknowledgements

Chung is supported by the Elsa Pardee Foundation Award, the Clinical and Translational Science Institute at Children's National Fund, the George Washington Cross-Disciplinary Research Fund, the George Washington Katzen Cancer Research Pilot Award. We thank M Sliwkowski, M Sagola, and Genentech, Inc for kindly arranging the transfer of an optical microscope and for previous scientific discussions. We are also grateful to A Schwartz, S Weiss, and A Chiaramello for their helpful feedback on the manuscript, I Teng for initial assistance with invasion assays, and P F G Rodriguez for assistance in setting up the nanoindenter. P Aswini provided valuable help with MS data analysis, and S Simmens offered helpful statistical consultation. N Suh contributed to discussions on MCF10A cell derivatives, and K Schill from DigitalScope assisted with IHC image data management. We also acknowledge H Yoo and C Chan for summarizing in vivo data and A Yohannes, H Yoo, J John, S Rachidi, S Ravulapalli, and R Yu for their efforts in cell management and raw image processing.

# Additional information

### Competing interests
Inhee Chung: is an inventor on US Patent 12013398 related to the findings described in this manuscript. The other authors declare that no competing interests exist.

### Funding

| Funder | Grant reference number | Author |
| --- | --- | --- |
| Elsa U. Pardee Foundation | | Inhee Chung |
| Children's National Hospital | The Clinical and Translational Science Institute | Inhee Chung |
| George Washington University | Cross-Disciplinary Research Fund, Katzen Cancer Research Pilot Award | Inhee Chung |

The funders had no role in study design, data collection and interpretation, or the decision to submit the work for publication.

### Author contributions
Xiangning Bu, Software, Formal analysis, Investigation, Visualization, Methodology; Nathanael Ashby, Investigation, Visualization, Writing – review and editing; Teresa Vitali, Sulgi Lee, Formal analysis, Investigation, Visualization; Ananya Gottumukkala, Sana Tabbara, Formal analysis; Kangsun Yun, Formal analysis, Visualization; Patricia Latham, Data curation, Formal analysis; Christine Teal, Data curation; Inhee Chung, Conceptualization, Resources, Data curation, Software, Formal analysis, Supervision, Funding acquisition, Validation, Investigation, Visualization, Methodology, Writing – original draft, Project administration, Writing – review and editing

### Author ORCIDs
Inhee Chung https://orcid.org/0000-0002-2312-6357

### Ethics
A retrospective clinical study was conducted to collect patient tissue blocks under IRB approval from George Washington University (IRB# NCR203065), with patient consent waived. Once the tissue specimens and associated clinical data for the study are identified, a code will be assigned to de-identify the patient specimens and clinical information for further study and analysis. The code links will be kept in a password-protected file in a secure location, physically separated from the database where the data was collected. Further use and analysis of tissues and clinical data will only be done using the assigned code.

Reviewer #1 (Public review): https://doi.org/10.7554/eLife.100490.4.sa1
Reviewer #2 (Public review): https://doi.org/10.7554/eLife.100490.4.sa2
Author response https://doi.org/10.7554/eLife.100490.4.sa3

# Additional files

### Supplementary files
MDAR checklist

### Data availability
Experimental data generated or analyzed during this study are included in the manuscript and supporting files.

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
