## [Editor Report · eLife Assessment]

This **fundamental** study provides **compelling** evidence that TRPV4 plays a crucial role in mechanical sensing during cancer cell transition from non-invasive to invasive states, and offers novel insights into metastasis. By employing multiple experimental approaches, including pharmacological and genetic manipulation, as well as advanced imaging techniques, the authors demonstrate a strong correlation between TRPV4 dynamics, calcium homeostasis, and cell volume plasticity. The findings significantly enhance our understanding of mechanotransduction in cancer and present TRPV4 as a promising therapeutic target for inhibiting metastasis.

---

## [Referee Report · Reviewer #1 (Public review)]

Summary:

In this study, Bu et al examined the dynamics of TRPV4 channel in cell overcrowding in carcinoma conditions. They investigated how cell crowding (or high cell confluence) triggers a mechano-transduction pathway involving TRPV4 channels in high-grade ductal carcinoma in situ (DCIS) cells that leads to large cell volume reduction (or cell volume plasticity) and pro-invasive phenotype.

In vitro, this pathway is highly selective for highly malignant invasive cell lines derived from a normal breast epithelial cell line (MCF10CA) compared to the parent cell line, but not present in another triple-negative invasive breast epithelial cell line (MDA-MB-231). The authors convincingly showed that enhanced TRPV4 plasmamembrane localization correlates with high-grade DCIS cells in patient tissue samples. Specifically in invasive MCF10DCIS.com cells they showed that overcrowding or over-confluence leads to a decrease in cell volume and intracellular calcium levels. This condition also triggers the trafficking of TRPV4 channels from intracellular stores (nucleus and potentially endosomes), to the plasma membrane (PM). When these over-confluent cells are incubated with a TRPV4 activator, there is an acute and substantial influx of calcium, attesting the fact that there are high number of TRPV4 channels present on the PM. Long-term incubation of these over-confluent cells with the TRPV4 activator results in the internalization of the PM-localized TRPV4 channels.

In contrast, cells plated at lower confluence primarily have TRPV4 channels localized in the nucleus and cytosol. Long-term incubation of these cells at lower confluence with a TRPV4 inhibitor leads to the relocation of TRPV4 channels to the plasma membrane from intracellular stores and a subsequent reduction in cell volume. Similarly, incubation of these cells at low confluence with PEG 3000 (a hyperosmotic agent) promotes the trafficking of TRPV4 channels from intracellular stores to the plasma membrane.

Strengths:

The study is elegantly designed and the findings are novel. Their findings on this mechano-transduction pathway involving TRPV4 channels, calcium homeostasis, cell volume plasticity, motility and invasiveness will have a great impact in the cancer field and potentially applicable to other fields as well. Experiments are well-planned and executed, and the data is convincing. Authors investigated TRVP4 dynamics using multiple different strategies- overcrowding, hyperosmotic stress, pharmacological and genetic means, and showed a good correlation between different phenomena.

---

## [Referee Report · Reviewer #2 (Public review)]

The metastasis poses a significant challenge in cancer treatment. During the transition from non-invasive cells to invasive metastasis cells, cancer cells usually experience mechanical stress due to a crowded cellular environment. The molecular mechanisms underlying mechanical signaling during this transition remain largely elusive. In this work, the authors utilize an in vitro cell culture system and advanced imaging techniques to investigate how non-invasive and invasive cells respond to cell crowding, respectively.

The results clearly show that pre-malignant cells exhibit a more pronounced reduction in cell volume and are more prone to spreading compared to non-invasive cells. Furthermore, the study identifies that TRPV4, a calcium channel, relocates to the plasma membrane both in vitro and in vivo (patient's samples). Activation and inhibition of TRPV4 channel can modulate the cell volume and cell mobility. These results unveil a novel mechanism of mechanical sensing in cancer cells, potentially offering new avenues for therapeutic intervention targeting cancer metastasis by modulating TRPV4 activity. This is a very comprehensive study, and the data presented in the paper are clear and convincing. The study represents a very important advance in our understanding of the mechanical biology of cancer.

---

## [Author Response]

The following is the authors’ response to the previous reviews.

**Public Reviews:**

**Reviewer #1 (Public review):**
Summary:In this study, Bu et al examined the dynamics of TRPV4 channel in cell overcrowding in carcinoma conditions. They investigated how cell crowding (or high cell confluence) triggers a mechano-transduction pathway involving TRPV4 channels in high-grade ductal carcinoma in situ (DCIS) cells that leads to large cell volume reduction (or cell volume plasticity) and proinvasive phenotype.In vitro, this pathway is highly selective for highly malignant invasive cell lines derived from a normal breast epithelial cell line (MCF10CA) compared to the parent cell line, but not present in another triple-negative invasive breast epithelial cell line (MDA-MB-231). The authors convincingly showed that enhanced TRPV4 plasmamembrane localization correlates with highgrade DCIS cells in patient tissue samples. Specifically in invasive MCF10DCIS.com cells they showed that overcrowding or over-confluence leads to a decrease in cell volume and intracellular calcium levels. This condition also triggers the trafficking of TRPV4 channels from intracellular stores (nucleus and potentially endosomes), to the plasma membrane (PM). When these over-confluent cells are incubated with a TRPV4 activator, there is an acute and substantial influx of calcium, attesting the fact that there are high number of TRPV4 channels present on the PM. Long-term incubation of these over-confluent cells with the TRPV4 activator results in the internalization of the PM-localized TRPV4 channels.In contrast, cells plated at lower confluence primarily have TRPV4 channels localized in the nucleus and cytosol. Long-term incubation of these cells at lower confluence with a TRPV4 inhibitor leads to the relocation of TRPV4 channels to the plasma membrane from intracellular stores and a subsequent reduction in cell volume. Similarly, incubation of these cells at low confluence with PEG 3000 (a hyperosmotic agent) promotes the trafficking of TRPV4 channels from intracellular stores to the plasma membrane.Strengths:The study is elegantly designed and the findings are novel. Their findings on this mechanotransduction pathway involving TRPV4 channels, calcium homeostasis, cell volume plasticity, motility and invasiveness will have a great impact in the cancer field and potentially applicable to other fields as well. Experiments are well-planned and executed, and the data is convincing. Authors investigated TRVP4 dynamics using multiple different strategies- overcrowding, hyperosmotic stress, pharmacological and genetic means, and showed a good correlation between different phenomena.All of my previous concerns have been addressed. The quality of the manuscript has improved significantly.

We are deeply grateful to the reviewer for their thoughtful assessment and invaluable suggestions, including crucial additional experiments and more effective presentation and description of our findings, which have greatly enhanced the quality of our manuscript.

**Reviewer #2 (Public review):**
Summary:The metastasis poses a significant challenge in cancer treatment. During the transition from non-invasive cells to invasive metastasis cells, cancer cells usually experience mechanical stress due to a crowded cellular environment. The molecular mechanisms underlying mechanical signaling during this transition remain largely elusive. In this work, the authors utilize an in vitro cell culture system and advanced imaging techniques to investigate how non-invasive and invasive cells respond to cell crowding, respectively.The results clearly show that pre-malignant cells exhibit a more pronounced reduction in cell volume and are more prone to spreading compared to non-invasive cells. Furthermore, the study identifies that TRPV4, a calcium channel, relocates to the plasma membrane both in vitro and in vivo (patient's samples). Activation and inhibition of TRPV4 channel can modulate the cell volume and cell mobility. These results unveil a novel mechanism of mechanical sensing in cancer cells, potentially offering new avenues for therapeutic intervention targeting cancer metastasis by modulating TRPV4 activity. This is a very comprehensive study, and the data presented in the paper are clear and convincing. The study represents a very important advance in our understanding of the mechanical biology of cancer.

We sincerely appreciate the reviewer’s insightful evaluation and invaluable recommendations for key additional experiments, which have significantly strengthened our manuscript.